

# Weather-induced crop failure events under climate change: a storyline approach

Henrique M.D. Goulart[1,2], Karin van der Wiel[3], Christian Folberth[4], Juraj Balkovic[4], and Bart van den Hurk[1,2]

[1]Deltares, Delft, The Netherlands
[2]Institute for Environmental Studies, VU University Amsterdam, The Netherlands
[3]Royal Netherlands Meteorological Institute (KNMI), De Bilt, The Netherlands
[4]International Institute for Applied Systems Analysis (IIASA), Ecosystem Services and Management Program, Laxenburg, Austria

**Correspondence:** Henrique Goulart (henrique.goulart@deltares.nl)

**Abstract.** Unfavourable weather is a common cause for crop failures all over the world. Whilst extreme weather conditions may cause extreme impacts, crop failure commonly is induced by the occurrence of multiple and combined anomalous meteorological drivers. For these cases, the explanation of conditions leading to crop failure is complex, as the links connecting weather and crop yield can be multiple and non-linear. Furthermore, climate change is likely to perturb the meteorological conditions, possibly altering the occurrences of crop failures or leading to unprecedented drivers of extreme impacts. The goal
of this study is to identify important meteorological drivers that cause crop failures and to explore changes in crop failures due to global warming. For that, we focus on a historical failure event, the extreme low soybean production during the 2012 season in the Midwest US. We first train a random forest model to identify the most relevant meteorological drivers of historical crop failures and to predict crop failure probabilities. Second, we explore the influence of global warming on crop failures and on
the structure of compound drivers. We use large ensembles from the EC-Earth global climate model, corresponding to present day, pre-industrial + 2 °C and 3 °C warming respectively, to isolate the global warming component. Finally, we explore the meteorological conditions inductive for the 2012 crop failure, and construct analogues of these failure conditions in future climate settings. Unlike present-day conditions, future warming may increase the probability of crop failures resulting from univariate meteorological features, reducing the importance of compound failure drivers. Impact-analogues show a signifi-
cant increase under global warming, with changes in the corresponding drivers. This has implications for risk assessment, as changing drivers of extreme impact events are highly relevant.

*Keywords*: climate change, crop failure, compound events, machine learning, storylines, large ensemble



## 1 Introduction

Soybeans are important for modern global society. They are used for human consumption, the main source of protein for animal feed worldwide and the second most consumed type of vegetable oil (Hartman et al., 2011). The vast majority of its production is concentrated in specific regions in Argentina, Brazil and United States of America, accounting for 80% of the world production (Hartman et al., 2011; Maria et al., 2020). The difference in scale between local production and global consumption makes soybeans the most traded crop in value in the world (FAO, 2021). Such a broad and extensive trade network renders the soybean supply chain especially vulnerable to local perturbations at the growing regions. Local shocks on production sites can potentially have worldwide consequences, as evidenced by the 2012 season, when exceptional low yields in most of the Midwest of the United States (Figure 2) drove global soybean prices to the highest values ever recorded (Zhang et al., 2018).

Weather and climate events have direct influence on agricultural production (IPCC, 2012). On a global level, interannual climate variability is responsible for around 30% of year-to-year crop variations(Lobell and Field, 2007), with specific regions above 60% of yield variability explained by climate variability (Ray et al., 2015; Frieler et al., 2017). Extremes weather events are also linked to crop failures (Vogel et al., 2019). In addition, unprecedented weather conditions due to anthropogenic global warming may alter crop failure frequency and the climatic drivers behind failures. Recent warming trends are already impacting crops worldwide in multiple ways and further warm conditions are expected to exacerbate these impacts (Schauberger et al., 2017; Moore and Lobell, 2015; Ray et al., 2019; Zhao et al., 2017; Wolski et al., 2020; Iizumi and Ramankutty, 2016; Zhu and Troy, 2018).

While extreme weather events, such as abnormally low levels of precipitation or excessive heat, can alone cause disruptions of crop development (Deryng et al., 2014), the majority of climatic shocks are compound events (Zscheischler et al., 2017; Zampieri et al., 2017). Compound events are combinations of multiple climate drivers that lead to an extreme impact, without necessarily being extreme themselves (Leonard et al., 2014; Zscheischler et al., 2018). Compound events should ideally be assessed from an impact perspective or the analysis should at least account for the complexity of weather-impact relations, rather than relying exclusively on extreme weather states (Zscheischler and Fischer, 2020; van der Wiel et al., 2020). Dealing with this complexity requires the use of explicit models (van den Hurk et al., 2015; van der Wiel et al., 2020), and a common alternative is to use statistical models to represent compound events. From linear models (Ben-Ari et al., 2018; Vogel et al., 2020) to deep neural networks (Crane-Droesch, 2018), statistical models in climate studies have been successful to link extreme impacts to weather and to explain specific unusual events.

Explaining individual events is part of the event attribution domain. It aims to determine both the influence of random weather and the footprint of climate change in individual cases (Trenberth et al., 2015; van Oldenborgh et al., 2021). Storylines of climatic events that lead to high impacts may be used to explore complex events, related drivers and interactions, improving risk awareness and strengthening decision making (Shepherd et al., 2018). Storylines start from a given impact, be it historical or physically plausible, and create a physically-sound chain of events from the impact to the driving components (Shepherd et al., 2018). The advantages of this approach are to quantify and understand the driving components and the influence of



climate change, and also the possibility of perturbing the driving components for the creation of analogues. Analogues are alternative realisations of a reference event that are perturbed by hypothetical conditions. Storylines can naturally embed the complexity of compound events and offer a framework to explore future analogues under different global warming scenarios
(Shepherd, 2019). Since crop failures are compound events, storylines can be built from a historical crop season of interest in order to disentangle the driving components and to generate analogues of the historical season under influence of climate change.

There are many recent studies that explore the interactions between crop and climate (Gawęda et al., 2020; Zipper et al., 2016; Heino et al., 2018; Iizumi et al., 2014; Zampieri et al., 2017; Ogutu et al., 2018). Some have included the possible impacts
of global warming under different scenarios (Rosenzweig et al., 2014; Lobell and Tebaldi, 2014; Feng et al., 2019; Xie et al., 2018). Others aimed to represent the compound nature of crop failures (Ben-Ari et al., 2018; van der Wiel et al., 2020; Vogel et al., 2020; Hamed et al., 2021; Zhu et al., 2021). van der Wiel et al. (2020) show the complexity between climate and crops by explicitly modelling the full distribution of climate impacts on crops with a physical crop model and large ensembles of climatic data. They demonstrate that links between extreme weather and extreme impacts are non-linear and the need for modelling
impacts. Vogel et al. (2020) apply a statistical linear model to automatically identify the most relevant meteorological variables for simulated extreme impact events in large ensemble crop data. They conclude that compounding effects are ubiquitous across time and meteorological drivers for crop failures. Hamed et al. (2021) use a statistical linear model to identify dominant within season climatic drivers that influence soybean yield variability in the US and highlight the synergistic effects between summer heat and moisture conditions modulating the final impact on yields. They find that, in spite of beneficial summer wetting and
cooling in the Midwest region largely attributed to agricultural intensification, the frequency of damaging joint hot and dry conditions remains largely unchanged. Ben-Ari et al. (2018) also apply a statistical linear model to successfully link climatic conditions with crop failures, including the identification of an extreme season that was not detected by the existing forecast models. Moreover, they analyse univariately the trends for each of the selected meteorological variables for different levels of global warming. Building on these works, a natural next step is to explicitly model the compound nature of meteorological
variables and their interactions, and to use the model to explore crop failures under different global warming scenarios.

In this study we explain historical soybean failures, explore possible future analogues and assess changes in the compound drivers due to rising temperatures. The work is divided in three parts (Figure 1): first, we develop a statistical model that links soybean failures generated by a crop model to local meteorological conditions. We use a non-linear and non-parametric statistical model (random forest) that accounts for compound drivers and that allows interpretation of the driving conditions
(Figure 1a). Second, we apply the model to 6000 years of climate data under different scenarios of global warming for failure analysis (Figure 1b). Third, we evaluate analogues of the 2012 season in the global warming scenarios using two different approaches (Figure 1c). Details on features selection, model training and the setup for the future analogues, along with the data used for this work, are presented in Section 2. The selected features are show in Section 3.1, while the performance of the model and the explanation of the driving components are demonstrated in Section 3.2. The use of large ensembles for
global warming scenarios and the role of compound events for crop failures are found in Section 3.3, while the exploration



of analogues of the 2012 season is shown in Section 3.4. The findings are put into context and debated in Section 4, and a summary of the work with its main messages is presented in Section 5.

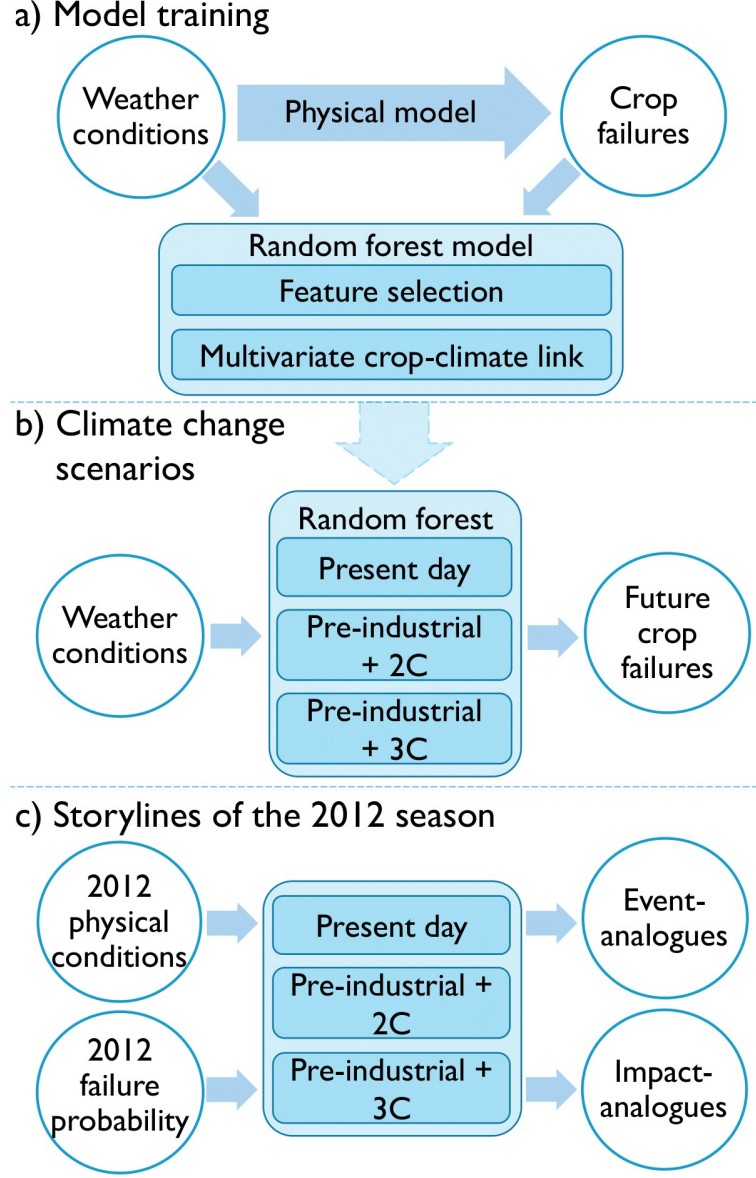

**Figure 1.** Experimental outline for this work. a) Model training: The process of training a random forest model to link multiple local meteorological variables to crop failures. b) Climate change scenarios: The extension of the trained random forest model for global warming scenarios to predict future soybean failure ratios. c) Storylines of the 2012 season: Construction of analogues to the 2012 season using two different approaches: the event-analogues and the impact-analogues.



## 2 Data and methods

### 2.1 Weather and crop data

We constructed a random forest model that identifies relationships between crop development and meteorological variables during the growing season. For crop data, we adopted yearly soybean yields (ton/ha of dry matter) generated by the global gridded crop model (GGCM) EPIC-IIASA (Balkovič et al., 2014), which is based on the Environmental Policy Integrated Climate (EPIC; Williams et al., 1995) field-scale crop model. This GGCM simulates complex relations between weather conditions and crops at planetary scales, by reproducing biophysical processes in the soil-plant-atmosphere system and providing

crop-related outputs based on climatic-related inputs. Simulation outputs used in this study were performed for phase 3a of the Intersectoral Impact Model Intercomparison Project (ISIMIP; see https://isimip.org for details and protocols) and the Global Gridded Crop Model Intercomparison (GGCMI) initiative. It used as climatic inputs the GSWP3-W5E5 dataset, which is a merge between the GSWP3 (Global Soil Wetness Project phase 3) dataset (Dirmeyer et al., 2006) and the W5E5 dataset (Lange, 2019). It captures the period from 1901 to 2016 at a 0.5° x 0.5° resolution. The reasons for using yields from crop

models are the longer timeseries that allow for more years to be included in the training of the statistical model and uniformity of data quality for regions where there is low quality of observational data. Other works have also adopted simulated yields for climate-crop analysis (Vogel et al., 2020; Zhu et al., 2021). Details of the EPIC-IIASA model's performance can be seen in Müller et al. (2017). We used 100 years of data (1916 until 2016) to train the model and we limited the analysis to grid cells from the top ten US soybean producer states, which are (ordered by production volume): Illinois, Iowa, Minnesota, Indiana,

Nebraska, Ohio, South Dakota, North Dakota, Missouri and Arkansas. Together, they represent over 70% of the US national soybean production and approximately 20% of the global soybean production (FAO, 2021). In addition, to ensure only rainfed soybeans were considered, we selected only grid cells that contained at least 90% of the grid as rainfed, which corresponded to 84% of the region studied (MIRCA2000; Portmann et al., 2010).

For weather data, we used observed data obtained from the Climate Research Unit (CRU) TS4.04 dataset (Harris et al.,

2020). It has global coverage at a resolution of 0.5° x 0.5, covers the period from 1901 to 2019, and is based on weather station observations. The CRU dataset has a comprehensive range of climatic variables at monthly resolution (Table 1), which makes the dataset suitable for agricultural studies, as shown in Kent et al. (2017); Zhu and Troy (2018); Vogel et al. (2019); Hamed et al. (2021).

Because the analysis involves also the exploration of future scenarios of global warming (GW), we included large ensembles

of synthetic weather data produced by the global climate model (GCM) EC-Earth V2.3 (Hazeleger et al., 2012). As a global coupled climate model, it combines atmospheric, ocean, land surface and sea ice models at a resolution of 1.125° x 1.125°. We used large ensembles of short-time periods to represent the full range of possible realisations at different levels of global warming (Van der Wiel et al., 2019). Three scenarios are considered: a benchmark representing the present day climate of 2011-2015 (PD), a similar 5-year period at 2 °C global mean temperature warmer than the pre-industrial levels (2C) and another

5-year period at 3 °C above pre-industrial levels (3C). To create the large ensembles for each GW scenario, we combined the 5-year periods with 16 different initial conditions and 25 different realisations based on stochastic physics. Together, they





| Variable name | Description | Units |
|:---:|:---:|:---:|
| Tmp | Mean temperature | °C |
| Tmn | Minimum temperature | °C |
| Tmx | Maximum temperature | °C |
| Dtr | Diurnal temperature range | °C |
| Precip | Precipitation amount | mm/month |
| Wet | Wet days per month | days |
| Vap | Vapour pressure | hPa |
| Pet | Potential evapotranspiration | mm/day |
| Frs | Ground frost days | days |
| Cld | Cloud cover | % |

**Table 1.** Meteorological variables and their descriptions

culminate in 2000 years of different simulations for each warming level scenario (Figure C1, see Van der Wiel et al. (2019) for more information on the ensemble setup). Finally, we resampled the large ensembles in 20 members of 100 years to be consistent with the length of the historical dataset (referred to as grouped ensembles hereinafter).

## 2.2    Data aggregation and detrending

To facilitate the comparison of observed and modelled data, we first upscaled the CRU and crop model data to the same resolution of the EC-Earth data with the first order conservative remapping method (Jones, 1999). We spatially averaged all data for the region studied (Figure 2) to focus on the regional scale of weather events and their crop yield impacts. The individual grid cells show a high spatial correlation with their neighbours and the local scale of impacts are not meaningful

for national and global implications. By aggregating the data for the entire region, the regional climatic conditions and crop failures are focus of study.

The setup of this work relies on comparing large uniform samples at different levels of global warming. Therefore, long term trends in the data needed to be removed. The yield data from the EPIC-IIASA model has a significant long term trend, which is a result of the inclusion of atmospheric $CO_2$ concentration levels in the biomass growth calculation, a process called $CO_2$

fertilisation (Deryng et al., 2016; Toreti et al., 2020). We regressed the yield data against the global $CO_2$ concentration levels to remove the long term trend (Figure C2). Here we explore the probability of soybean failure, which is defined by means of a threshold, where every season with yield one standard deviation below the mean was considered a failure, similarly to (Ben-Ari et al., 2018; Zhu et al., 2021). The meteorological variables from the CRU dataset were also detrended linearly to remove global warming influence. This way, we isolated the interannual variability component from long-term trends in both

meteorological variables and crop yield timeseries.





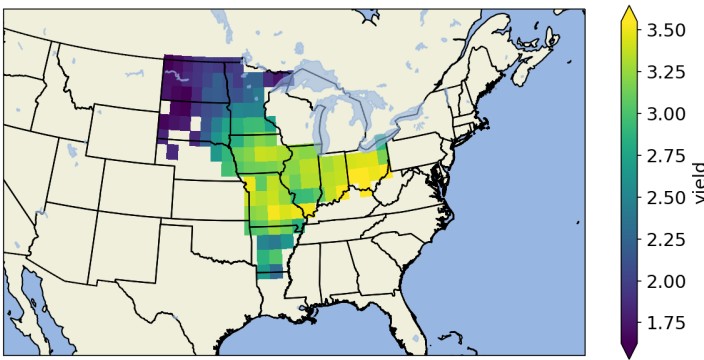

**Figure 2.** Selected grid points for the main producer states and the mean yields (ton/ha) per grid cell.

## 2.3 Training and validation of the random forest model

We chose a random forest model to detect failures in soybean yields because of its high performance, flexibility and interpretability. Random forest (Breiman, 2001) is a non-linear and non-parametric statistical model for classification and regression. The model consists of an ensemble of independent decision trees. The decision trees are each trained on random

sub-samples of the data to provide different predictions, and the final estimate of the model takes into consideration all predictions together, accounting for internal variability. Random forest has become widely popular and is applied in different fields of science. It presents high accuracy while providing low overfitting levels (Breiman, 2001) and is ranked among the best classifiers for real world problems (Fernández-Delgado et al., 2014).

The first step in designing the random forest model was the feature selection. Among the multiple meteorological variables

considered (Table 1), some variables are more relevant than others in predicting crop failures for the region studied. There is also a temporal factor, where the importance of a meteorological variable shows a seasonal cycle. By removing non-relevant variables and months, the data fed to the random forest model is simplified and the model performance possibly increased. We considered multiple feature selection methods because there is no universal method, providing robustness to the selection. The methods are: analysis of variance (ANOVA) (Anderson, 2001), mutual information selection (Kraskov et al., 2004), chi-

squared test, and the internal feature selection of the random forest model (Breiman, 2001). At the end of the feature selection step, we obtained the most important meteorological variables to soybean failures.

Random forest models require the tuning of internal parameters for an optimal performance. We tuned the random forest's parameters following a cross-validation of 10 different data splits and 5 random reshuffles. Its configuration can be seen in Table C3. Because the crop yield dataset has less failure seasons than non-failure seasons, the dataset is imbalanced. To

address this issue and improve the model performance, we assigned weights that were inversely proportional to the frequency of each class. Whilst non-parametric and non-linear models like random forests tend to achieve higher performance than





simpler linear models, their complexity renders it more difficult to interpret the outcomes. We included partial dependence plots in the analysis. Partial dependence plots illustrate how model outputs vary according to alterations in one or more inputs, while preserving other inputs values (Friedman, 2001). These plots make the random forest model more interpretable and
demonstrate the interactions between meteorological variables and crop yields.

In order to evaluate the random forest performance, we used the Mathews correlation coefficient (MCC) metric. It is considered more informative and truthful for the evaluation of binary classification models than other metrics (Chicco and Jurman, 2020). The MCC assesses the performance of the model by quantifying the number of true positives (TP), the number of true negatives (TN), the number of false positives (FP) and the number of false negatives (FN), as illustrated in Equation 1. It ranges
from -1 to 1, and a score of 0 is equivalent to a random prediction. The model requires both positive data and negative data to be correctly predicted to have a high score, which makes it particularly useful for imbalanced datasets (Chicco and Jurman, 2020). For a comprehensive overview of the model performance, we included additional performance metrics, which are: accuracy, precision, recall and f1-score. They can be seen in Appendix A.

$$MCC = \frac{TP*TN - FP*FN}{\sqrt{(TP+FP)(TP+FN)(TN+FP)(TN+FN)}} \qquad (1)$$

We compare the random forest's performance to threshold conditioned methods, which are frequently used for multivariate risk assessment (Serinaldi, 2016; Salvadori et al., 2016; Zscheischler and Seneviratne, 2017). Two cases are here adopted: the "AND" and the "OR" cases. The "AND" case requires all variables to be equal or above hazard limits simultaneously for the failure definition. The "OR" case considers at least one of the conditions surpassing the limit to classify as failure (Salvadori et al., 2016). The threshold values for this work were defined as the average conditions of failure seasons in the observed data
for each variable minus the corresponding standard deviation of that variables across the sample of failure seasons.

### 2.4 Exploration of global warming scenarios

To explore soybean failures at different levels of global warming, we used large ensembles of meteorological data from the EC-Earth model. The large ensembles have the advantage of explicitly simulating extreme events that would not be found in smaller datasets due to their rare nature (Van der Wiel et al., 2019). Because the RF model relies on thresholds that may be exceeded
more or less frequently in biased climate model data, bias corrections were applied for the analysed regions. We estimated adjustment factors between the detrended CRU dataset between 1916 and 2016 and the PD scenario. Bias adjustment factors were calculated with Detrended Quantile Mapping (Cannon et al., 2015), considering 25 degrees of freedom. The adjustment factors were then applied to the scenarios 2C and 3C, assuming a constant bias (see Figure C3 for bias correction results for each scenario and Figure C4 for spatial variability of corrected bias).
To assess the importance of compound events, we created permuted versions of each large ensemble by randomly reshuffling the meteorological variables, so that the correlation structure between them was removed (referred to as shuffled versions). We also defined a metric that quantifies the importance of multivariate processes to crop failure (in contrast to single-variate processes), the compound factor. It is the ratio of the failure ratio obtained with the original data to the failure ratio obtained





with the shuffled data, $Compound\ factor = \frac{Failure\ ratio_{original}}{Failure\ ratio_{shuffled}}$. Therefore, for scenario exploration, we use six scenarios:
PD, PD-shuffled, 2C, 2C-shuffled, 3C and 3C-shuffled, each of which was fed into the RF model to obtain the soybean failure
probabilities. We assessed the impact of climate change by comparing the failure probabilities for different return periods
(calculated as the inverse of the failure probability) for the PD, 2C and 3C scenarios. Then, we quantified the compound factor
by comparing each level of global warming with their shuffled variants. Finally, we compared the RF model with the threshold
conditioned methods to account for differences in the approaches to predict changes with global warming and to quantify the
compound factor.

Machine learning algorithms do not extrapolate well for data outside the training range (Hengl et al., 2018). Given the
random forest model in this work is trained on historical conditions but applied to GW scenarios, we tested the influence of
data outside the training range in the analysis following a three-step test: 1) Identify the number of cases outside the training
range for each meteorological variable; 2) Restrict the values outside the training range to be within the training range; 3)
Quantify possible differences and inconsistencies in the results between the data before and after the conversion of values
outside the training data range (see Appendix B for more information).

## 2.5  Development of storylines

Storylines were here used to identify the driving components of a historical extreme event and to generate future analogues
based on the same event. The 2012 soybean season is our case study, which presented extremely low yields across the main
producing regions of the United States. We first verified whether the trained RF model was able to correctly predict 2012
as a failure event and the failure probability assigned to it. To explore the possible analogues of the 2012 season in warmer
scenarios, we took two approaches based on the event definition: 1) the first approach was based on the physical conditions
that lead to the 2012 extreme event, defined here as "event-analogues". We quantified the joint occurrences of meteorological
conditions exceeding the 2012 season conditions in the global warming (GW) scenarios; 2) the second approach was based on
the impact metric of the event, its yield failure probability estimated by the random forest model, and we defined this approach
as "impact-analogues" (as e.g. in Van der Wiel et al., 2019). Using this approach, we quantified the number of seasons in
the GW scenarios with equal or higher failure probability predicted by the random forest model. Last, we compared the 2012
season meteorological conditions with the mean meteorological conditions of impact-analogues to account for possible changes
in the physical aspects of the future analogues.

## 3  Results

### 3.1  Feature evaluation and selection

Following the methods section, the first step to build a random forest model is the feature selection. The four feature selection
methods (Table C1) show potential evapotranspiration, diurnal temperature range, monthly maximum temperature and precipi-
tation as the most important features. Furthermore, we see a strong seasonal influence, with the most important variables taking





place in July and August. The two months correspond to the reproductive phase of soybeans in the region studied (Bastidas et al., 2008; Hatfield et al., 2018), which is a vulnerable crop stage to weather stress (Hatfield et al., 2011; Siebers et al., 2015; Hatfield and Prueger, 2015; Hamed et al., 2021). Based on this, we test an alternative version of the meteorological variables limited to July and August and aggregated along the two months. Results show the aggregated data outperforms the monthly data (Table C2). We therefore adopt for this study the aggregated version of the meteorological variables along July and August

and compare them to their climatology to identify general features (Figure 3). Failure years are warmer, have lower levels of precipitation and less wet days, larger daily temperature range, higher levels of potential evapotranspiration and lower fractions of cloud cover when compared to the climatology. Vapour pressure and ground frost frequency do not show significant differences between normal and failure seasons, so they are removed from the model training.

The remaining meteorological variables have high correlation levels, as seen in Figure 3b. To separate which correlations

have statistical redundancy and which have not, we examine each meteorological variable individually. Among the monthly temperature variables, mean (tmp), minimum (tmn) and maximum (tmx) temperature variables are highly interconnected. Tmx shows the best performance among the three variables, so we select tmx as the representative of temperature variables. Monthly precipitation and wet days per month are also highly interconnected, but their performance is similar. As precipitation is a precursor of wet days per month in the CRU dataset (Harris et al., 2020), monthly precipitation is selected. Potential

evapotranspiration exhibits high correlation values to all other variables because potential evapotranspiration is derived from temperature, vapour pressure and cloud cover (Harris et al., 2020). Therefore, we consider potential evapotranspiration redundant for this specific experiment. Finally, the three variables selected are maximum monthly temperature, precipitation and DTR. They still have considerable levels of correlation, but these have a physical meaning, highlighting the compound nature of meteorological variables leading to crop failures in the region.

## 3.2   Random forest model evaluation

To evaluate the performance of the random forest (RF) model to link meteorological variables to crop yield failures, we use the Matthews correlation coefficient (MCC) metric (eq. 1), and compare the results with the "AND" and "OR" threshold conditioned approaches for reference. The RF model has the highest MCC score at 0.61, while the "AND" approach has 0.54 and the "OR" method only 0.34. The random forest model also performs better than the other methods in the additional

metrics as seen in Figure C5. Therefore, the RF model is successful to link inputs with outputs and outperforms the threshold conditioned methods.

Partial dependence plots (Figure 4) explore which variables contribute to successfully distinguish between failure and non-failure conditions. They show the relationships between the meteorological variables' perturbations and crop failure probability. For the selected time period, crop failures are proportional to diurnal temperature range and maximum temperature. Precipi-

tation shows a general inverse proportion to crop failure probability, suggesting low values of precipitation to increase failure probability, but high values of precipitation also somewhat increase the failure probability, as indicated by Figure 4. Soybean failures in the region studied are thus associated to high levels of monthly maximum temperature, high levels of diurnal


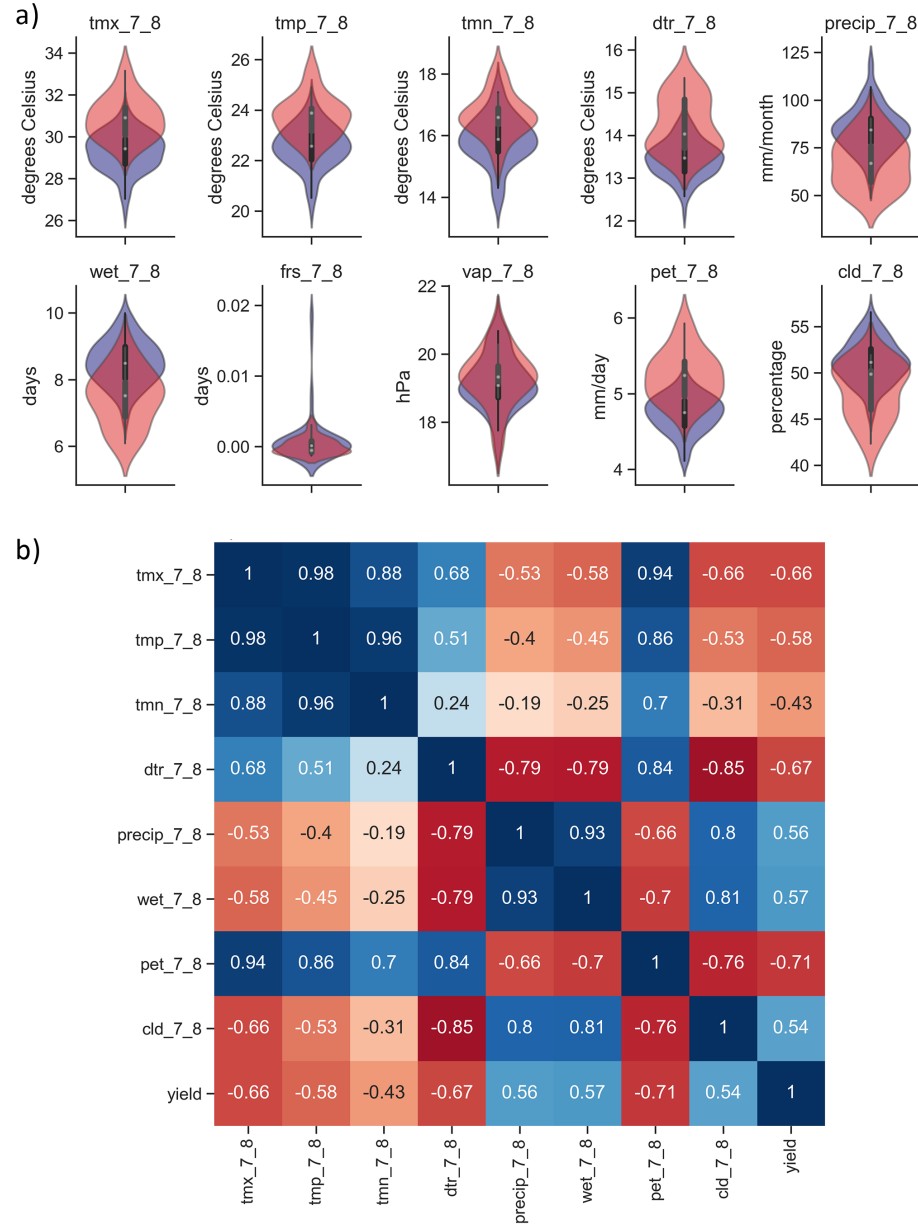

**Figure 3.** a) Probability distribution function of monthly values for meteorological conditions of soybean seasons with (red) and without (blue) failures for maximum monthly temperature (tmx), precipitation (precip), diurnal temperature range (dtr), vapour pressure (vap), potential evapotranspiration (pet) and cloud cover (cld). Numbers 7 and 8 indicate the months of July and August respectively. b) Pearson's correlation matrix indicating the correlation levels for meteorological variables aggregated along July and August, and crop yields.





temperature range, and to low levels of precipitation. Furthermore, we observe that the links between crop failures and the meteorological variables are non-linear.

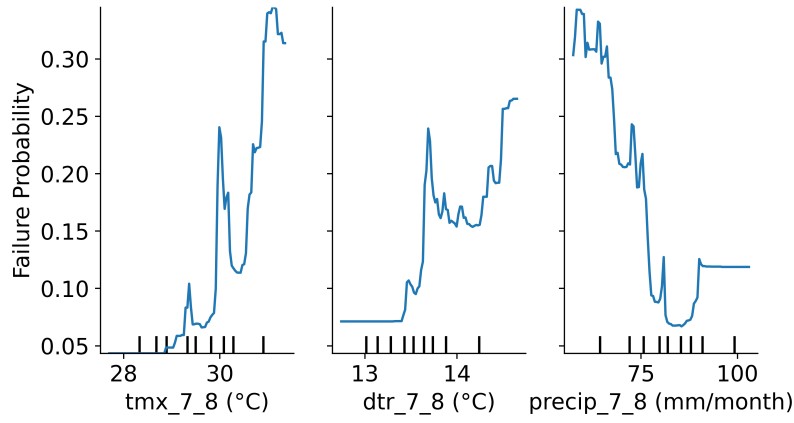

**Figure 4.** Partial dependence plots showing the failure probability (0 to 1.0) given the variation of the meteorological variables, monthly maximum temperature, diurnal temperature range and monthly precipitation along the months of July and August.

The random forest model manages to successfully link weather conditions with crop failures, to capture the complexity of the failure, including non-linear relationships, and to outperform the approaches "AND" and "OR". The success in reproducing crop failure impacts from combinations of weather features for the historical datasets makes the model suitable to explore different climate scenarios.

### 3.3    Scenario exploration

The trained and validated random forest model is applied to the grouped ensembles to estimate crop failure probabilities for different global warming scenarios. First we determine if the crop failure probabilities obtained for the present day (PD) scenario is consistent with the observed data (Figure 5a). The width of the grouped ensemble (shading) indicates the many possible manifestations due to natural variability. The observed data are just a single realisation over that period and we see the observed data are within the range of the ensemble. For the estimation of global warming influence on crop failure probabilities,

we quantify the return periods of failure probabilities for the PD, 2C and 3C scenarios (Figure 5b). The 2C scenario shows increased failure probabilities for any given return period with respect to the PD scenario, while the 3C scenario shows slightly higher failure probabilities than the 2C scenario. Global warming is therefore likely to increase the occurrence of soybean failures, but the difference between 2C and 3C is not significant.

The compound factor is quantified by considering two versions of data arrangement: original (ordered) and shuffled (un-

ordered, no correlation between variables). Figure 6 presents the crop failure probabilities for different return periods. In all scenarios, the return periods for the 0.5 failure probability threshold are shorter for the original data than for the shuffled data. Thus, the combination of the meteorological variables is relevant for the failure likelihood of soybeans (e.g.low precipitation





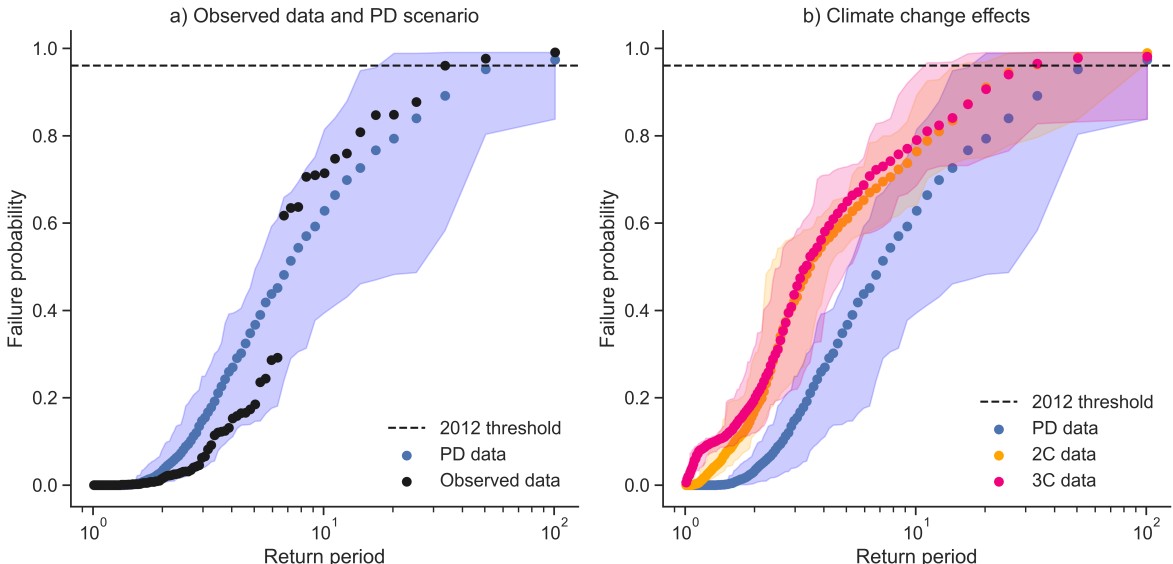

**Figure 5.** a) Random Forest model failure probabilities for different return periods, comparing observed data (black) to the PD scenario (blue). b) Return periods for PD (blue), 2C (orange) and 3C (pink). Dots show the 20 member mean, shading shows the range across the 20 members. The dashed line represents the failure probability of the 2012 season predicted by the RF model (Section 3.4).

concurs with above average temperatures), which highlights the compound nature of the crop failure drivers. For the 2000 years of data in each scenario, the RF model predicts 276 failure seasons for the PD original data whereas the shuffled version has

63 failure seasons predicted. The failure seasons for the 2C original and shuffled data are, respectively, 616 and 241 seasons, and for the 3C scenario, 621 and 353 seasons. Thus, while the number of failure seasons increases with global warming for the original data, the increase in the number of failure seasons for the shuffled data is higher. This indicates that univariate weather extremes more frequently lead to crop failures, decreasing the compound factor under global warming scenarios.

We demonstrate the changes in the physical conditions for failure and non-failure seasons at each scenario (Figure 7 for 3C

and Figure C6 for 2C). Median temperature values along July and August of failure years increase by 1.1 °C for 2C and 2.6 °C for 3C. Median precipitation values of failure years show little change, slightly increasing by 2.4 mm/month for 2C and 0.18 mm/month for the 3C compared to PD. Median diurnal temperature range values of failure years show a slight decrease of 0.33 °C and 0.36 °C for 2C and 3C scenarios, respectively. Despite median values of precipitation not changing significantly, we see an increase in extreme dry years for 2C and 3C (Figure 7a top). Together with temperature extremes, the results suggest

more frequent and intense joint warm and dry conditions in the future (Figure 7a). However, we see also an increase in the failure distribution at higher (and thus less critical) precipitation levels for both 2C and 3C than for PD. A similar pattern is observed for the diurnal temperature range values, where less critical levels of DTR still incur in failures (Figure 7b). Extreme temperature values dominate the failure probability at warmer levels, which reduce the need of other variables to be extreme as



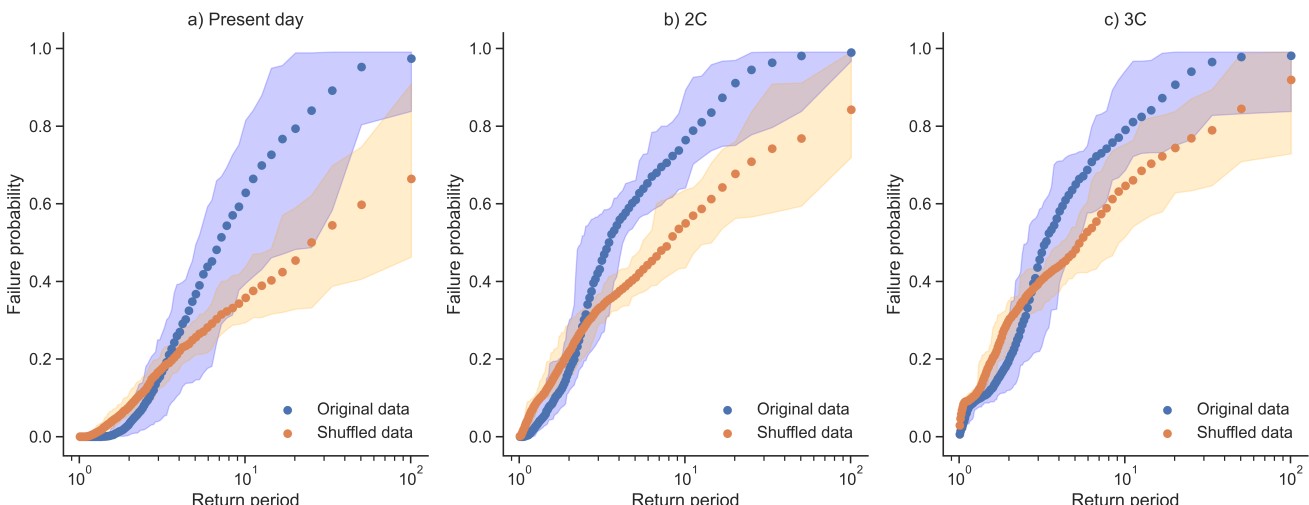

**Figure 6.** Random Forest model failure probabilities for different return periods, based on original data (variable correlations as normal, blue) and shuffled data (variable correlation removed, orange) for three climatic periods: a) present-day, b) pre-industrial +2C warming and c) pre-industrial +3C warming. Dots show the 20 member mean, shading shows the envelope of variability for the grouped ensemble.

well to generate failures. Therefore, the univariate increase in the temperature values due to global warming is associated with

the increase of soybean failure ratios and with the decrease of the compound factor.

We compare the RF model with the "AND" and "OR" approaches to account for differences in the approaches in quantifying the compound factor and the predicted changes with global warming. The "AND" approach shows the lowest ratio of failure for both original and shuffled data (Figure 8a), but also the highest levels of compound factor (the difference between ordered and permuted datasets, Figure 8b). The "OR" approach only requires one critical variable for failure definition, which implies the

highest failure ratios (Figure 8a). Moreover, breaking the correlation structure and shuffling the meteorological variables for this approach means increasing the number of failure seasons, which implies compound factor has a decreasing role. Finally, the random forest model predicts an intermediate number of failure seasons. The RF model suggests compound factor as an enhancing factor for crop failures (in contrast to the "OR" method), but not as much as suggested by the "AND" model (Figure 8a). Because the RF model presented the highest performance scores previously, we consider it to be the most reliable in

quantifying compound factor, while the others either underestimate or overestimate the importance of compound structure for crop failures. Nevertheless, all three methods agree that compound factor looses importance under warmer scenarios, trending towards a level of 1.0 (no compound factor, Figure 8b).

The extrapolation test indicates all three meteorological variables in the climate projections have values outside the training range, with temperature extremes exceeding the historical range as the most frequent case of extrapolation. However, the

conversion of values outside the training range into values within the training range do not change the results obtained. This is because the model is a binary classifier, and the failure is defined within the historical data. Unprecedented extreme values do





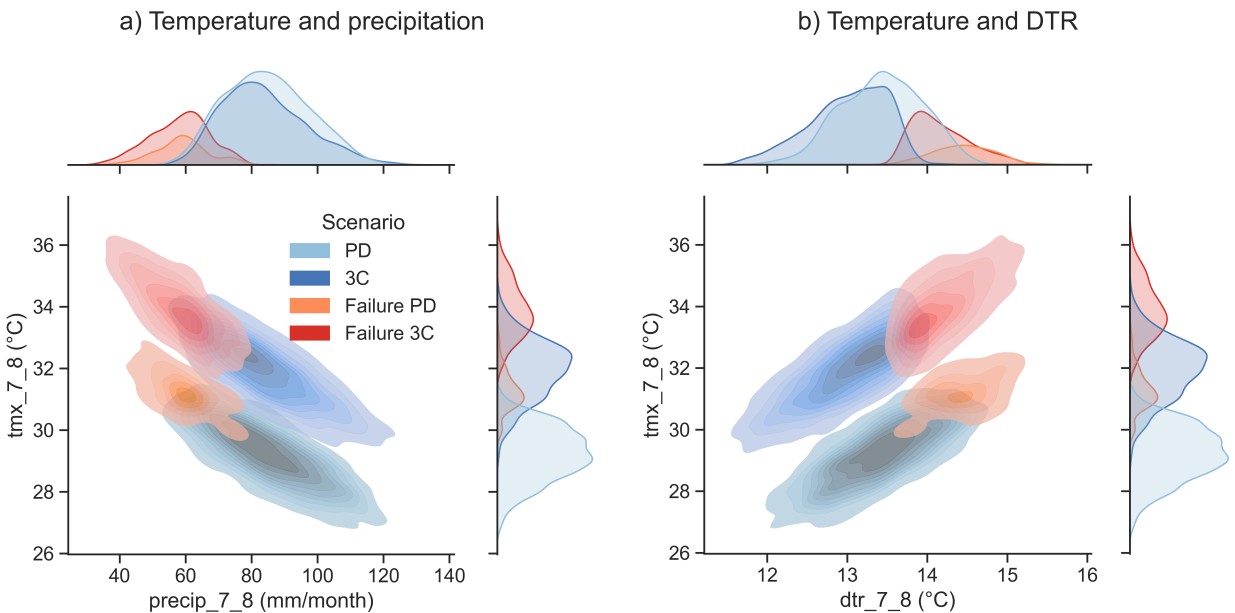

**Figure 7.** Kernel density estimate plots for seasons with and without failures at different GW scenarios: a) for maximum monthly temperature and precipitation and b) temperature and diurnal temperature range.

not change the failure definition, as it follows the assumption that values outside the training range are similar to the extremes in the training range (See Appendix B for further information).

With the extrapolation test demonstrating that values outside the training range do not affect the results, the impacts of different levels of climate change on soybean failure probability in the region are validated. Next we investigate if these general conclusions also hold for specific cases like the extreme 2012 season using a storyline approach.

### 3.4 Storyline analysis: the 2012 season and future analogues

The year 2012 had an extreme loss in US soybean production, with exceptionally low yields in the majority of the producing region (Figure 9a). We test the random forest model for the 2012 season to identify the meteorological variables associated with the failure event. The season stood out as all three selected meteorological variables presented extreme values (Figure 9b), with precipitation approximately at -2 STD (standard deviations), temperature above 2 STD, and DTR scoring the highest recorded value exceeding 3 STD. When used to predict the probability of crop failure due to the meteorological conditions described above, the RF model indicates a 0.96 likelihood of the 2012 season to be failure.

We create 2012 season analogues using both impact and event perspectives. Event-analogues are defined as the joint occurrences of the 2012 climatic conditions (as in Figure 9), while impact-analogues are defined as the number of seasons with failure probability assigned by the random forest model as equal or higher than the probability of the 2012 season (i.e. probability >= 0.96). The event-analogues have a occurrence ratio of 0.0015 (3 events in 2000 years, 666 years of return period) for

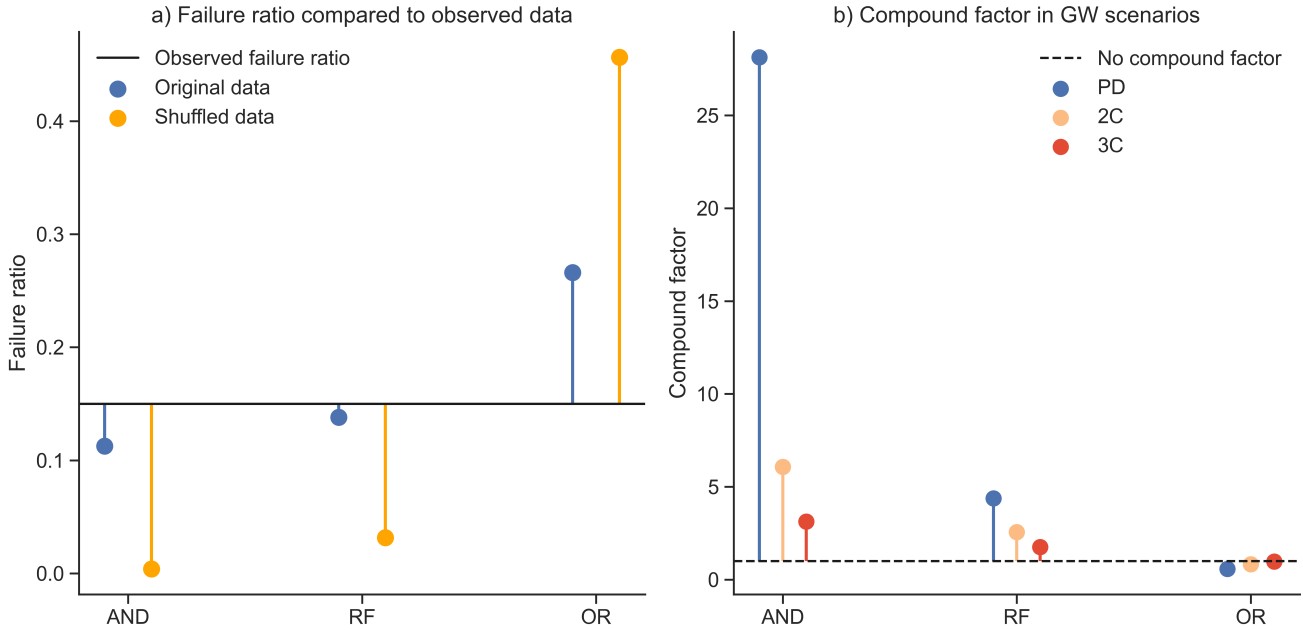

**Figure 8.** a) The failure ratios for ordered and shuffled PD data for the three approaches: AND, OR and RF with respect to the failure ratio of observed data; b) compound factor level (original failure ratio divided by shuffled failure ratio) of each approach for the GW scenarios.

both PD and 2C scenarios, but is not seen in the 3C scenario (Figure 10a), suggesting a highly rare event that does not increase with global warming. However, we find a higher ratio of 2012 impact-analogues for the PD scenario, 0.022 (44 years of return

period), 14 times the PD event-analogues. Impact-analogues also increase in frequency at warmer scenarios, 0.038 (26 years of return period) for 2C and 0.042 (24 years of return period) for 3C. The difference in results between the two types of analogues is mainly due to the DTR values, illustrated by Figure 10e, f, g, which shows the ratio of exceedance (the frequency of seasons exceeding a given threshold) of the 2012 meteorological conditions for each variable individually. Event-analogues require all variables to exceed the corresponding 2012 conditions. While we observe an increase in the number of seasons with tempera-

ture and precipitation exceeding the 2012 conditions in global warming scenarios, DTR does not increase with GW. Therefore, DTR becomes a bottleneck in the generation of event-analogues. Impact-analogues, on the other hand, are predicted to increase with global warming because they are defined based on the impact metric (failure probability) and bypass the DTR limitation. In addition, by relying on the impact metric, the meteorological conditions of the analogues can be analysed for changes due to global warming. Results show impact-analogues of 2012 at 2C and 3C scenarios are hotter, drier and have lower DTR values

than the original 2012 event and PD analogues (Figure C7).

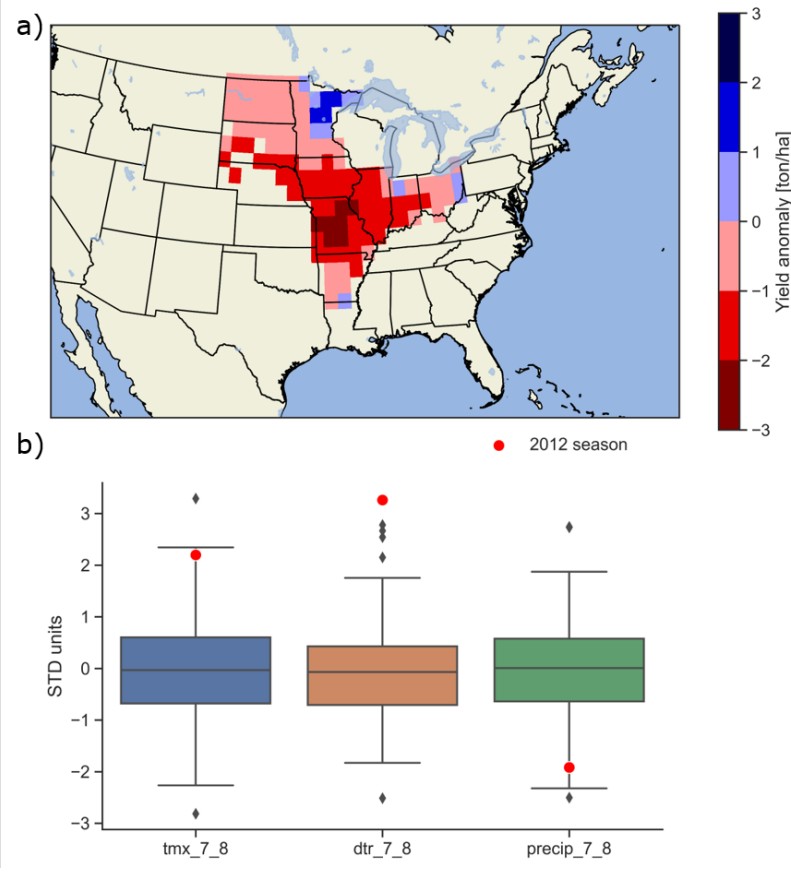

**Figure 9.** a) Map of the yield anomaly for the 2012 season compared to the averaged historical yield data; b) Normalised meteorological variables for the observed dataset and the corresponding 2012 season climatic conditions.

## 4 Discussion

A large portion of crop failures is attributed to combinations of meteorological drivers (Zampieri et al., 2017; Zscheischler et al., 2017) and the interactions between weather and crops are known to be nonlinear (Schlenker and Roberts, 2009; Zscheischler et al., 2017). It is a challenge, therefore, to identify the variety of combinations of weather conditions that can lead to crop

failures. We argue that a model explicitly designed based on the impact, here crop failure, allows the conversion of a multivariate problem into a univariate variable, which simplifies the analysis, and improves the quality of results (van der Wiel et al., 2020). The random forest model here developed is successful at predicting soybean failure seasons and shows an overall better performance than benchmark methods. It adds to the list of works that demonstrate the usefulness of impact-inspired approaches (Ben-Ari et al., 2018; Vogel et al., 2020; van der Wiel et al., 2020; Hamed et al., 2021; Zhu et al., 2021). The

feature selection process used here combines machine learning with findings from the literature to identify compound drivers of soybean failures in the US. Temperature, precipitation and diurnal temperature range along July and August are deemed key

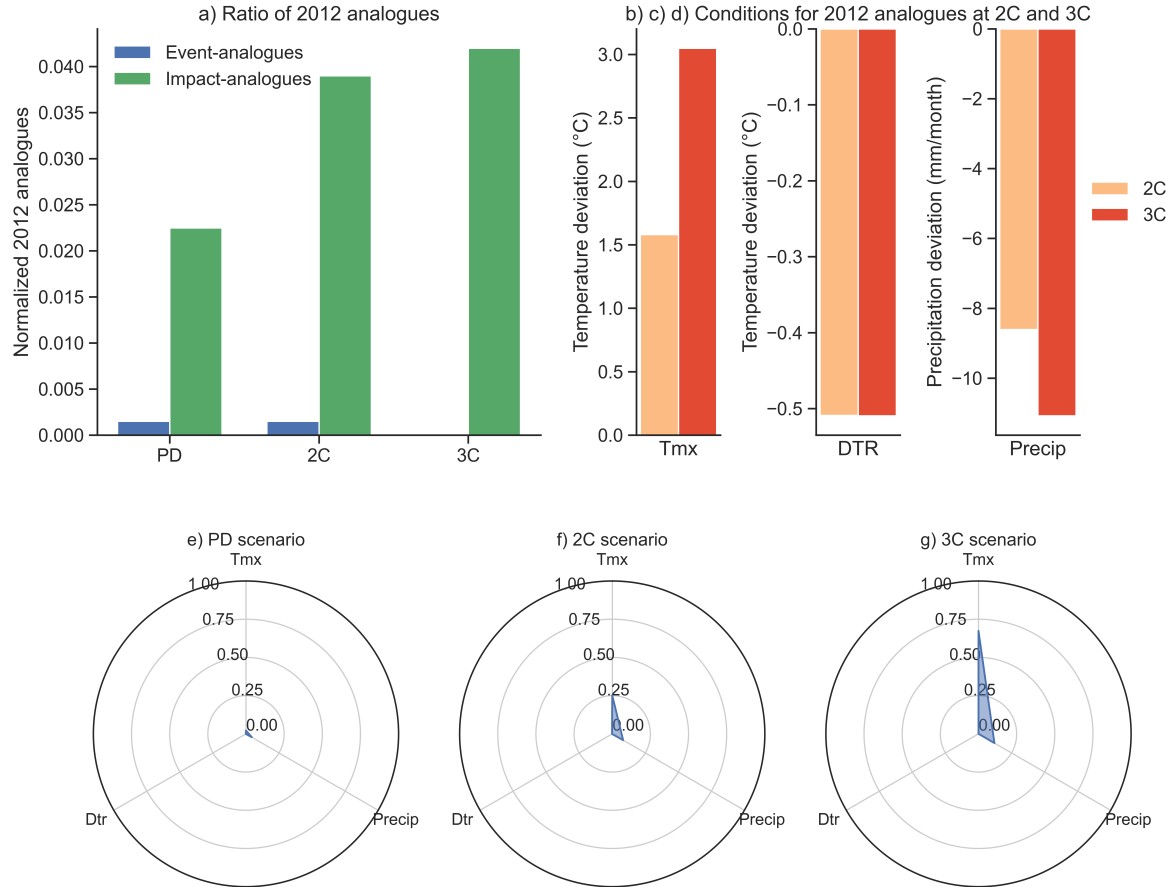

**Figure 10.** a) Event-analogues (blue) and impact-analogues (green) for the 2012 season; The deviation of climatic conditions between the 2012 event and the 2012 impact-analogues for maximum temperature (b), DTR (c) and precipitation (d). Radar graphs showing the number of seasons exceeding the 2012 values for each meteorological variables at PD (e), 2C (f) and 3C (g).

climatic drivers in the region studied. The meteorological variables identified in this work are in agreement with the work of Vogel et al. (2020), which also shows temperature, precipitation and DTR to be important meteorological variables for crop development, and with the work of Hamed et al. (2021), that highlights the harmful combination of hot and dry conditions

along summer for soybeans in the same region. This work considers only meteorological variables during the growing season of rainfed soybeans, so management practices, irrigation and sub-surface conditions are not considered. Another caveat is that we do not account for $CO_2$ concentrations, due to our focus on year to year variability. Yet, $CO_2$ fertilisation is an important factor when considering crop-climate interactions and when looking at the effects of future climate change (Schlenker and Roberts, 2009; Deryng et al., 2016; Toreti et al., 2020).





Even though extrapolation with statistical models carries risks (Hengl et al., 2018), there are works that train statistical models on historical data and apply them on global warming scenarios (Schlenker and Roberts, 2009; Roberts et al., 2017; Crane-Droesch, 2018; Zhu et al., 2021). The random forest model used here is a classifier trained to detect a class of failure events determined by the historical dataset. The decision trees within the random forest are threshold-based, so values outside the training range are categorised together with historical extremes. Furthermore, the purpose of the work is to measure the

frequency of historical failures and analogues under future conditions, instead of quantifying the magnitude of failures under unprecedented extreme conditions. Finally, we run a extrapolation test and results show no influence of values outside the training data range in the results (Appendix B). Together, these factors validate the application of the random forest model for global warming scenarios under these assumptions.

We use three large ensembles with different levels of global mean surface temperature to investigate the influence of climate
change on crop failure probabilities. Soybean failures increase in frequency for both 2 °C and 3 °C warmer worlds. When compared to the literature, some works support future global warming negatively affecting soybeans in the Unites States (Schlenker and Roberts, 2009; Deryng et al., 2014; Schauberger et al., 2017; Zhao et al., 2017), while others indicate that soybeans in the same region could actually benefit from climate change due to an intensification of local rainfall (Lesk et al., 2020). In the EC-Earth runs, we see a drying trend for the Midwest US (Figure 9d), which explains the different results.

Dynamical aspects of global warming have large uncertainties due to high internal variability (Fischer et al., 2014) and the ensemble of CMIP6 models do not show significant changes for the months of July and August in the region studied (Almazroui et al., 2021). The adoption of different global climate models and the evaluation of climate change impacts conditioned on storylines of different levels of precipitation are alternatives that could explain the discrepancies seen above and should be further explored in the future. Furthermore, we do not consider potential adaptation measures by farmers such as the expansion

of irrigation systems to counteract any possible drying trends.

Our results show climate change is expected to impact both univariate and multivariate components of crop failure. From an univariate perspective, temperature is expected to increase significantly and precipitation to decrease slightly, contributing each to more frequent failures. On the other hand, diurnal temperature range values are expected to decrease slightly, which would lower failure probability. DTR is relevant for agriculture, because it indicates peaks in high day-time temperature and

in low night-time temperature, which are both dangerous to crop developments (Vogel et al., 2019). The decrease in DTR values due to global warming is mainly associated with a higher increase of minimum night-time temperature (Qu et al., 2014; Sun et al., 2019). A hypothetical temperature peak under global warming conditions would imply in a lower DTR value than the same peak under present conditions. However, from a biophysical perspective, both peaks are equally harmful for soybeans. Hence, DTR becomes less suitable under global warming scenarios to indicate crop failures. From a multivariate

perspective, the compound factor of crop failure events is shown to decrease with global warming. Assuming no change in the cultivar properties, higher temperature distributions suggest the biophysical limits of a healthy development of soybeans are more frequently exceeded by temperature values alone, independently of the other meteorological conditions. The correlation between the contributing meteorological variables becomes less relevant, as evidenced by the scenario 3C (Figure 6c). This way, when paired with high temperature levels, non-critical values of precipitation and DTR can still lead to failures.



The random forest model defines the 2012 year as a highly likely failure season. That year, the climatic conditions during
the reproductive phase of the crop were dry, hot and had a particularly high diurnal temperature range. When considering the
event-analogues, i.e. looking at the event given its meteorological specifics, results show the 2012 season to be particularly
rare and unlikely to increase in frequency due to climate change in the projections considered herein. Yet, when assessing
the impact-analogues, defined as the likelihood of events with a similar failure probability using the random forest model,

more events are identified in the PD scenario compared to the event-analogues. Furthermore, we find a significant increases
are predicted for warmer scenarios. The differences between event-analogues and impact-analogues reflect the concept be-
hind each approach. Event-analogues are based on the physical conditions of the event and obtained by quantifying the joint
occurrence of these conditions in the ensembles. Impact-analogues, on the other hand, are based on the impact metric, ac-
counting for all combinations of weather that lead to the same failure probability of the 2012 season. This difference between

an event-based and impact-based approach was also highlighted by (van der Wiel et al., 2020). In spite of the exceptionally
high DTR value for the 2012 season, the GW scenarios show decreasing DTR values with respect to the mean temperature
increase. This evidence is supported by the literature, suggesting DTR is inversely proportional to global warming (Qu et al.,
2014; Sun et al., 2019). Lower DTR values therefore constrain the increase of 2012 event-analogues, even though the other
two meteorological variables show a significant increase. Since the random forest is able to explicitly account for the impact of

the 2012 season, it detects other possible combinations of meteorological variables leading to 2012 impact-analogues. Further-
more, impact-analogues of the 2012 season in the future see a change in their physical properties: they become warmer, drier
and with lower values of diurnal temperature range. Results highlight the importance of the inclusion of impact-analogues in
storylines creation. Storylines have been commonly used to generate counterfactuals by reproducing similar physical events to
historical ones, adopting an event-inspired perspective (Shepherd, 2019; Sillmann et al., 2020). However, storyline counterfac-

tuals (Shepherd et al., 2018) could profit from also explicitly considering the impact perspective, as we have shown with the
impact-analogues created with the random forest model to directly model crop failures. The inclusion of the impact perspective
into storylines allows for a more comprehensive view of future realisations when compared to considering only the occurrence
of similar physical events. Note that for society it is the impacts that matter, rather than the meteorological condition. Further-
more, using impact-analogues allows for the estimation of possible changes in the physical characteristics of analogues due to

global warming, increasing the robustness of climate risk assessment for future scenarios.

## 5   Conclusion

This work presents an evaluation of the impacts of global warming on weather-induced soybean failure events. Its novelty lies
in combining a statistical model capable of simulating non-linear and compound interactions, the random forest model, with
large ensembles of future global warming scenarios. The steps to create the model are the selection of the most important

meteorological variables during the growing phase of the crop and the training of the model on historical data. We explore the
influence of global warming on soybean crop failure with the use of large ensembles at different levels of global warming. The





model is successful at identifying failure seasons in the historical data and selecting the most relevant meteorological variables for crop failures. When compared to two benchmark methods, the model presents an overall better performance.

The main findings of the paper suggest soybean failures in the Midwestern United States are likely to increase with global warming due to warmer and drier atmospheric conditions. Global warming is likely to change both univariate and multivariate drivers of failure, decreasing the importance of compound factor in soybean failures. Estimations of future analogues of the 2012 season diverge according to the approach used. If considering the event-inspired threshold conditioned method, event-analogues are deemed extremely rare and do not increase in likelihood with global warming. However, when using the impact-inspired model here developed, we show that impact-analogues are actually more common than what was originally predicted by the threshold conditioned method, and the number of analogues is expected to significantly increase with warmer climates. Moreover, we observe changes in the physical properties of the impact-analogues under global warming, becoming warmer and drier, but with lower diurnal temperature range levels. Impact-analogues complement event-analogues, improving the risk estimation of storylines.

**Appendix A: Additional performance metrics**

To provide robustness to the performance analysis, additional performance metrics were used to evaluate the model. Accuracy quantifies the amount of true positives (TP) and true negatives (TN) out of the total data (that is the true data plus false positives (FP) and false negatives (FN) (eq. A1). Precision defines the fraction of true positives cases out of the total predicted positive cases by the model (eq. A2). Recall measures the correct fraction of positive cases out of the true positive cases (eq. A3). The F1 score is a metric used to address false positives and negatives (eq. A4).

$$Accuracy = \frac{TP+TN}{TP+TN+FP+FN} \tag{A1}$$

$$Precision = \frac{TP}{TP+FP} \tag{A2}$$

$$Recall = \frac{TP}{TP+FN} \tag{A3}$$

$$F1 = \frac{2TP}{2TP+FP+FN} \tag{A4}$$

**Appendix B: Extrapolation test**

Machine learning algorithms have the drawback of not extrapolating well for data outside the training range. However, there are works that use the extrapolation technique to predict crop impacts under global warming scenarios with statistical models





(Schlenker and Roberts, 2009). Yet, the setup of the experiment is to quantify the number of failure events in global warming scenarios based on failures already identified in the historical data. The presence of values outside the training data range might not affect the random forest model, because the same failure probabilities are assigned to these values as to the closest

values in the training range (corresponding maximum and minimum values). This is due to the decision trees that compose the random forest. The decision trees divide the input space using thresholds, which group similar input values together. Values outside the training data are grouped together with the corresponding extreme values within the training data, because they are within the same input space according to the decision tree (lower or higher than a threshold value). We verify the validity of the experiment with an extrapolation test based on three steps: 1) Identify the number of cases outside the training range for

each meteorological variable; 2) Convert the values outside the training range into the closest values within the training range; 3) Quantify differences and inconsistencies in the results between the GW scenarios before and after the conversion.

The number of events in each GW scenario that is outside the training range is shown in Table B1. After the conversion of values outside the training data range into values within the training data range, the results stayed the same for all scenarios (Figure B1). We assume the presence of values outside the training range not to influence the results obtained with the random

forest model.

| | PD Above | PD Below | 2C Above | 2C Below | 3C Above | 3C Below |
|---|---|---|---|---|---|---|
| tmx_7_8 | 2 | 0 | 103 | 0 | 607 | 0 |
| dtr_7_8 | 3 | 14 | 3 | 34 | 0 | 97 |
| precip_7_8 | 5 | 26 | 13 | 48 | 10 | 91 |

**Table B1.** Number of years with values above the historical maximum ("Above") and below the historical minimum ("Below") for each of the GW scenarios PD, 2C and 3C. Each scenario has 2000 years in total.





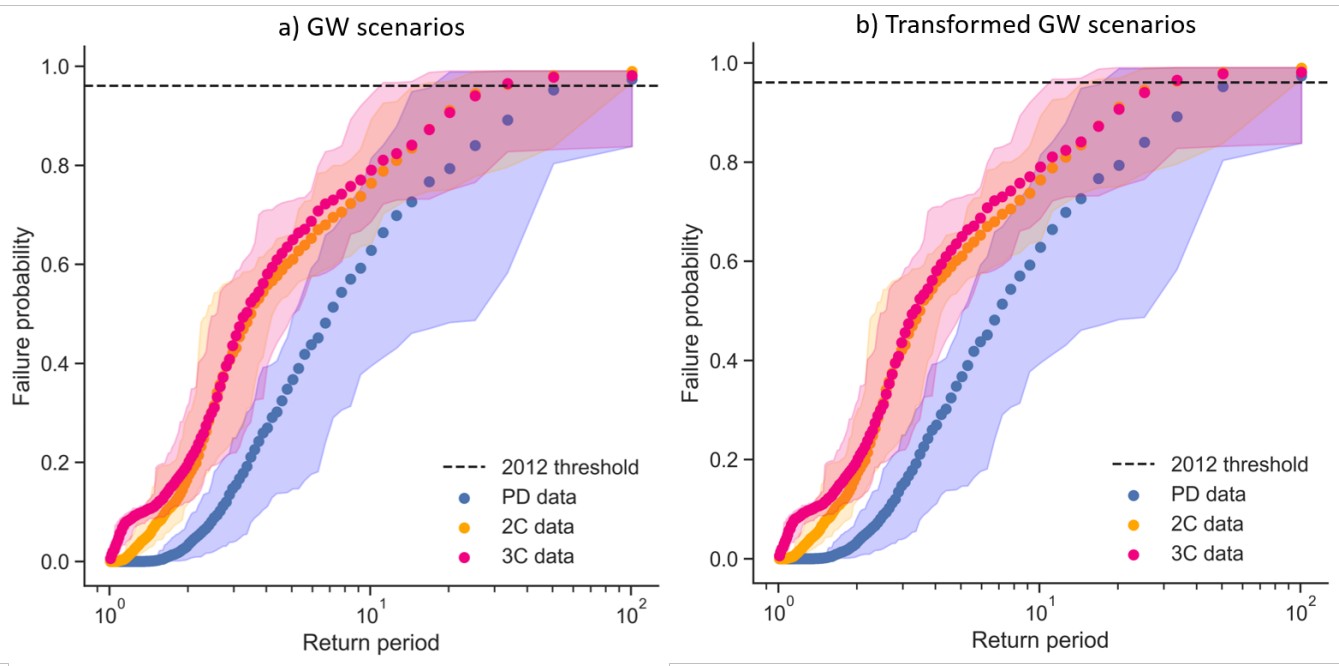

**Figure B1.** a) Random Forest model failure probabilities for PD (blue), 2C (orange) and 3C (pink), b) same as a) but for values adjusted to the training data limits. Dots show the 20 member mean, shading shows the range across the 20 members. The dashed line represents the failure probability of the 2012 season predicted by the RF.





**Appendix C: Supplementary figures**

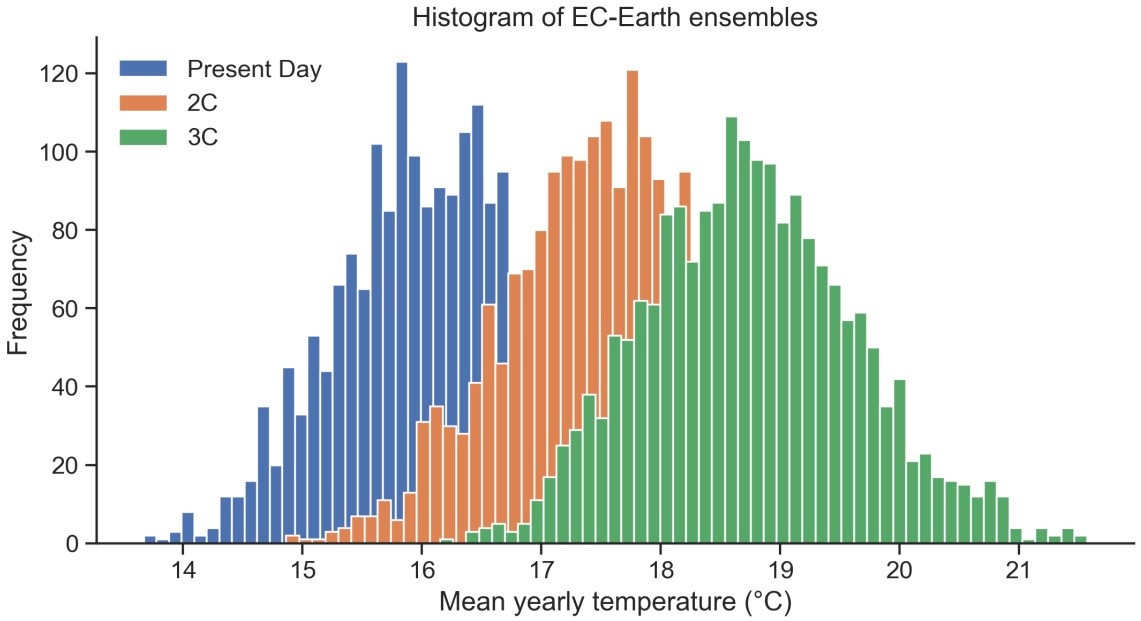

**Figure C1.** The distribution of mean yearly temperatures in the Midwestern US of EC-Earth ensembles for PD (blue), 2C (orange) and 3C (green) scenarios.

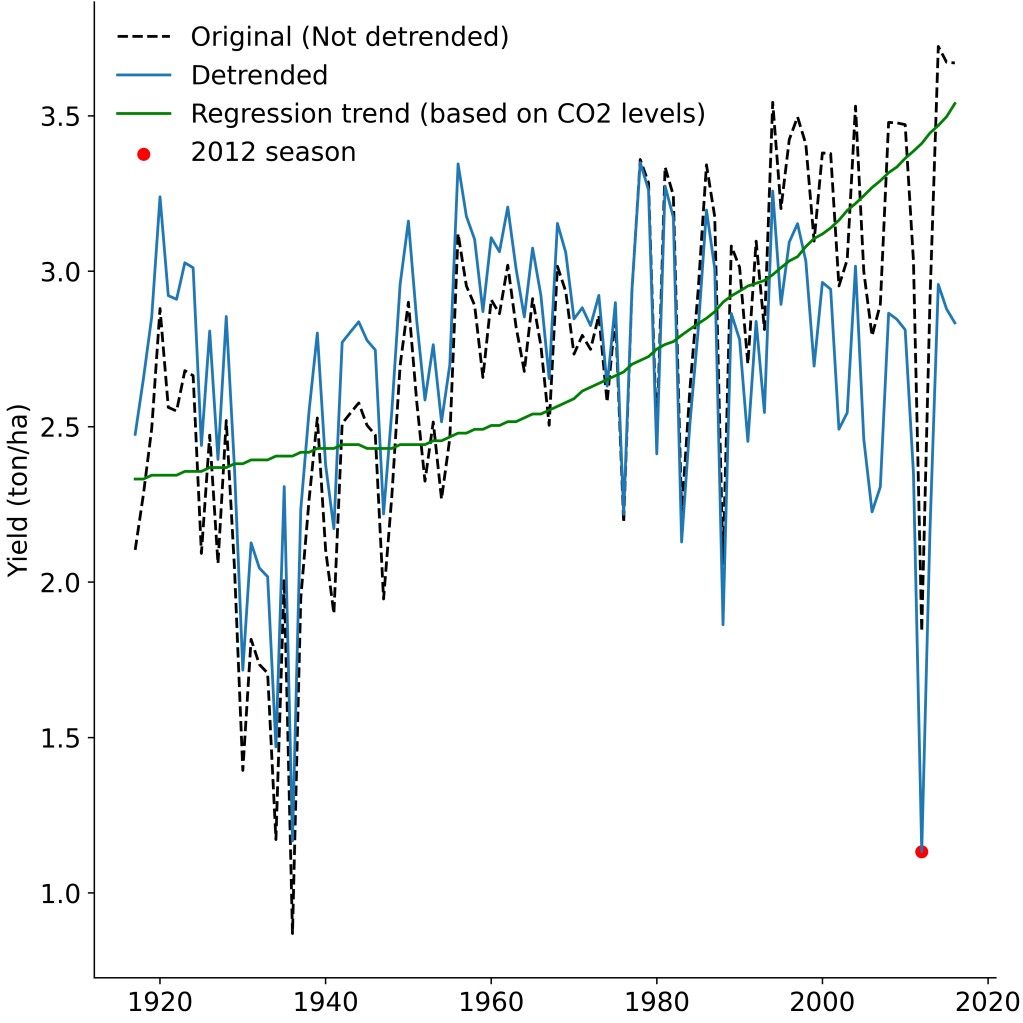

**Figure C2.** Timeseries of averaged US soybean yields for the selected area (black dotted line) and its detrended counterpart (blue). The 2012 season is highlighted in red. The regression trend based on the global $CO_2$ levels that is used to detrend the soybean yields is shown in green.





**Figure C3.** Mean annual cycle for the selected variables: Tmx (upper row), dtr (middle row) and pre (bottom row). Left column presents the observed data (black), the original EC-Earth PD data (blue) and the EC-Earth after bias correction (dashed red). On the central and right columns the bias corrected EC-Earth data for PD (dashed red) is then compared with the bias corrected versions of 2C (central columns, orange) and 3C (right column, green) scenarios.





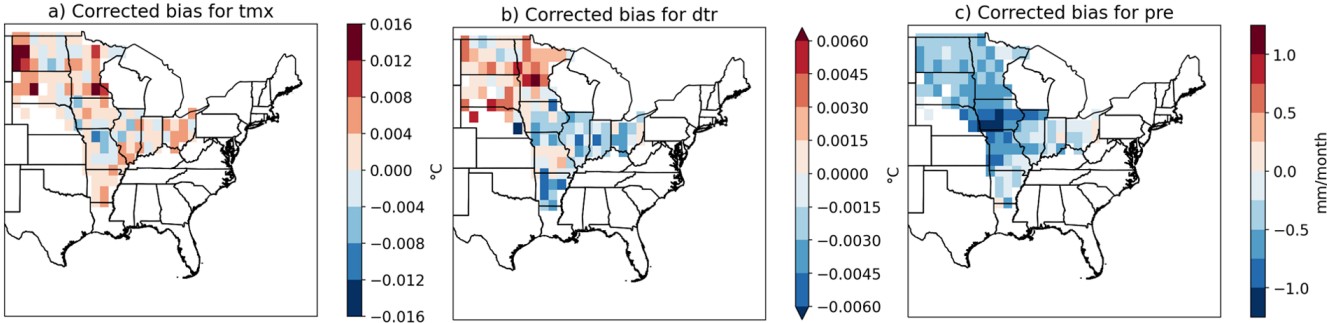

**Figure C4.** Spatial variability of bias corrected values between observed data and EC-Earth PD scenario for a) Temperature, b) Diurnal temperature range, c) Precipitation.





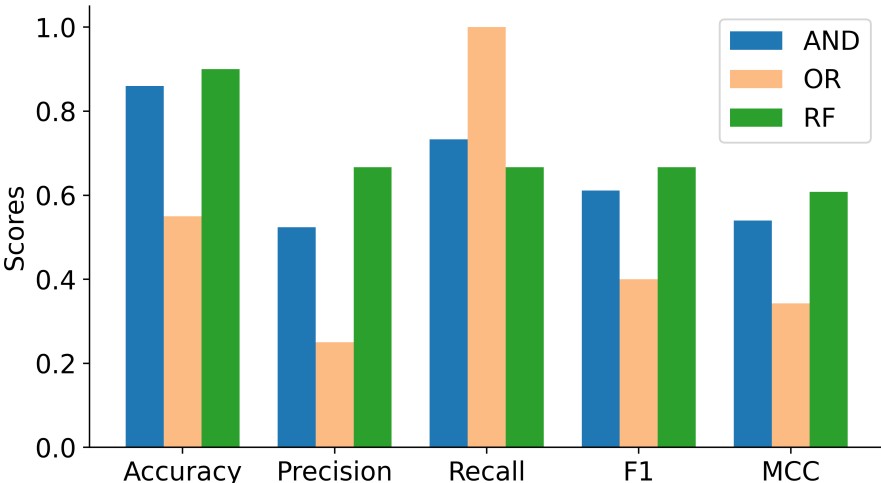

**Figure C5.** Evaluation of each approach in identifying crop failures for the observed data under different performance categories: Accuracy, Precision, Recall, F1 and MCC. The approaches are AND (blue), OR (orange) and RF (green).





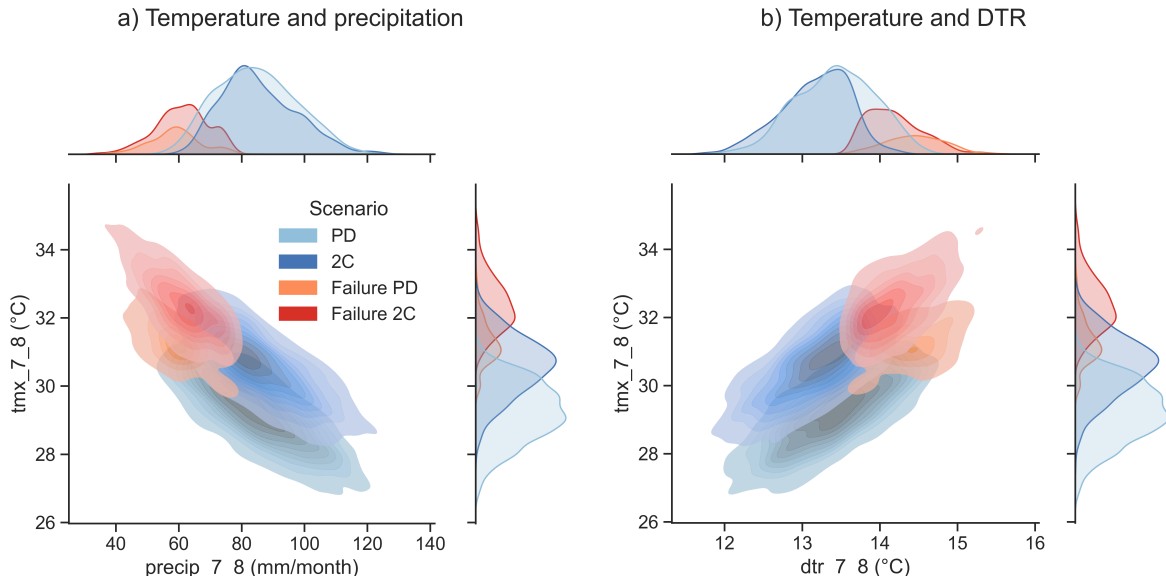

**Figure C6.** Same as Figure 7, but for the 2C scenario.





**Figure C7.** Same as Figure 7, but for the 2012 impact-analogues.





| Method | 1st | 2nd | 3rd | 4th | 5th |
|---|---|---|---|---|---|
| ANOVA | pet_8 | dtr_7 | pet_7 | tmx_8 | tmx_7 |
| Chi-2 | pet_8 | pet_7 | dtr_8 | dtr_7 | precip_7 |
| Mutual selection | pet_8 | tmx_8 | pet_6 | tmx_7 | dtr_7 |
| Random forest | dtr_7 | pet_7 | pet_8 | precip_7 | tmx_7 |

**Table C1.** Rank of the most important features for different feature selection methods along the entire soybean growing season: ANOVA, Mutual information selection, Random forest classsifier and Chi-2.





| Data arrengement | MCC | Accuracy | Precision | Recall | F1 |
|---|---|---|---|---|---|
| Monthly | 0.49 | 0.85 | 0.5 | 0.67 | 0.57 |
| Aggregated | 0.61 | 0.9 | 0.67 | 0.67 | 0.67 |

**Table C2.** Comparison of the random forest model performance between monthly data and aggregated data along reproductive phase of soybean growing season.



| Parameter | Description | Value |
|---|---|---|
| n_estimators | Number of trees in the model | 600 |
| max_depth | Maximum tree depth | 7 |
| max_features | Maximum features per split | Square root of total features |
| class_weight | Weights associated with classes | "Balanced_subsample" |

**Table C3.** Random forest model configuration of hyper-parameters.

*Code availability.* The code for this experiment is available at: https://github.com/dumontgoulart/agr_cli.

*Data availability.* CRU data is freely available with the cited literature. EC-Earth data is available with Karin van der Wiel (wiel@knmi.nl) on request. EPIC data is available at the Intersectoral Impact Model Intercomparison Project (ISIMIP).

*Author contributions.* HG, KvdV and BvdH contributed to the concept of the study. HG conduct the research and edited the manuscript. KvdV, CF and JB provided the data. All authors discussed the analysis and results. BvdH and KvdW supervised the work and revised the manuscript.

*Competing interests.* The authors declare that they have no competing interests.

*Special issue statement.* This article is submitted to the special issue "Understanding compound weather and climate events and related impacts".

*Acknowledgements.* This research has been supported by the European Union's Horizon 2020 research and innovation programme under grant agreement No 820712 (project RECEIPT, REmote Climate Effects and their Impact on European sustainability, Policy and Trade). The GSWP3-W5E5 climate dataset and growing season data were provided by the Global Gridded Crop Model Intercomparison (GGCMI) initiative and the Intersectoral Impact Model Intercomparison Project (ISIMIP). We thank D. Wagenaar, L. Leunge, R. Hamed, A. Alexander and I. Lombardich for comments on previous versions of the manuscript.



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
