# Peer review of "Storylines of weather-induced crop failure events under climate change"

_Earth System Dynamics, 2021_

## Author Comment (AC1)

**Weather-induced crop failure events under climate change: a storyline approach**

Henrique M.D. Goulart, Karin van der Wiel, Christian Folberth, Juraj Balkovic, and Bart van den Hurk

**= Response to the reviewers =**

**Reviewer #1:**

Goulart et al. provide an interesting analysis of soybean failures in the US as modeled using Random Forest in the present climate and in future climates. The research is well conducted and uses appropriate methods. My main concern is not with the methods or results, but with the interpretation of those results. The manuscript is well written and the figures describe the data well. I thank the authors for presenting the research in such a complete and coherent manner.

*We would like to genuinely thank the reviewer for their constructive review of our manuscript. In this document we respond to these comments and highlight the modification in the revised text/figures.*

**Major comments**

I rather like your use of Random Forest and you apply the model rigorously. Your use of shuffling data is also clever and provides a nice analysis structure, but your interpretation of the results could be improved. See my two comments below

- The interpretation of the compound factor becoming less important needs contextualization. I don't think it's quite correct to say that it becomes less important, but only that it becomes more dependent on precipitation. I think this relates to your test of importance being to shuffle the data in an attempt to break the correlation structure as compared to the ordered data. But in a warming climate, when all years are hot years, the dependence of joint hot-dry extremes becomes dependent mostly on rainfall, making the joint failure and shuffled failure years similar to one another at high return periods. I think it is misleading then to say that the compound nature of these events becomes less important, only that their occurrence depends primarily on precipitation anomalies.

*The reviewer makes a good point. First we decided to change the "compound factor" for "relative compound contribution" to avoid confusions. The reduction of the relative compound contribution is because more years are warm, and the definition of failure, as defined in the paper, becomes more dependent on rainfall. Therefore, in the future climate simulation the failure risk in the ordered data increases less than in the randomized data. We interpret it as follows:*
*- Statistically the reduction of the relative compound contribution shows the events are tending towards univariate behaviour, however*
*- From a biophysical perspective, the events are still characterised by a combination of hot and dry conditions.*

- L 429-438 - There is no real evidence (from figure 7) that precipitation contributes to the increase in soybean failures overall. There is only evidence that it contributes in a secondary way to truly extreme events, such as 2012 (Figure 10), although its influence is much smaller than that of temperature

*We agree with the reviewer. The distributions of precipitation for the warmer scenarios do not show large changes, perhaps a slightly drier tail. We updated the text to reflect this:*

*Line 469: "The main findings of the paper suggest soybean failures in the Midwestern United States are likely to increase with global warming mainly due to warmer atmospheric conditions during summer. With more frequent warmer years, the joint hot-dry conditions leading to crop failures become more common. Conversely, the increase in frequency of warmer years renders the occurrence of joint hot-dry extremes more dependent on summer precipitation anomalies, reducing the importance of the relative compound contribution. With global warming, the crop failures approximate statistically to a more univariate behaviour, despite still being physically the results of compound events."*

*Line 408: "In the EC-Earth runs, we see a drying trend for extreme years in the Midwest US (Figure 10d), which explains the different results."*

- There is an unintended consequence of using a threshold to define failures that leads to the conclusion that joint dependence will become less important in the future. When you set a threshold to identify "failure" years arising from hot years, you lose the sensitivity of yields to warming above that threshold. For example, in the future even if all years meet the criteria for hot years, drought years may be exceptionally hot. This joint temperature-precipitation correlation structure would show up if you were modeling yields as a continuous variable but is lost once you have converted your dependent variable to a binary (e.g. no discrimination between "failure" and "exceptional failure").
  - In particular, your analysis cannot say that "the correlation between contributing meteorological variables becomes less relevant" (L392-393). You can only say that it becomes less relevant for your chosen yield threshold. If you were to up your threshold definition of "failure", it would once again become important.

*The reviewer is correct in their point about events just above the threshold. In our analysis we have chosen to define crop failure events as those years with a yield below the historical standard deviation levels. We have updated the text to be more precise and explicit. A similar use of binary data, without qualification of the strength of failure, is quite common. It is, for example, used in Ben-Ari et al. (2018), Vogel et al. (2021) and Zhu et al. (2021).*

*Line 429: "... the correlation structure of the variables contributes to the occurrence of compound events, as previously shown by (van den Hurk et al., 2015) and (Santos et al., 2021). Yet, we observe a decrease in its importance under global warming conditions for the failure yield threshold adopted here (Figure 6)."*

  - It is not true that "non-critical values of precipitation [...] can lead to failures" (L394). We can see in Figure 7 that in fact all of the 3C failures still occur at very low levels of precipitation, but it is true that now low levels of precipitation lead to crop failures at higher rates. My qualm here is really with the interpretation of the precipitation not DTR as you already discuss the limitations of the model with respect to DTR well in the discussion section.

*This is true, precipitation is not changing much between the scenarios. We have updated the text.*

*Line 416: "... In a warmer world, the same levels of precipitation lead to higher rates of crop failures than under the current climate conditions. ... A higher frequency of years with critical temperature during summer makes crop failures mostly dependent on precipitation values. Therefore, while still physically a compound event, the soybean failures under global warming become statistically similar to a univariate event based on precipitation"*

**Minor comments**

Does EPIC-IIASA include changing management in the historical run or is it static?

*EPIC treats management as static. We have updated the text to make this more explicit.*

*Line 106: "... Another advantage of using simulated crop models is that management and technology trends are static, whereas they are intrinsically embedded in the observed yield datasets."*

Why do there only appear to be nine values in your partial dependence plots (Figure 4)? Or am I misinterpreting the ticks at the bottom of the plots on the x-axis?

*The ticks represent the decile marks (9 ticks to divide the space in 10). All reviewers found it confusing, so we have decided to remove them.*

Figure C7 - not required, but I'd suggest using different colors. It's nearly impossible to distinguish the differences in the PDFs at the top of the graph using orange/red and yellow

*Thank you for the suggestion, it has been updated.*

L 355 - I'd qualify that the change in temperature is much greater than the change in precipitation

*Would it be possible that there was a typo in the line number? We cannot link the statement to the text at line 355.*

L288-289 - It is not true that you see an increase in extreme dry years. The two blue PDFs are nearly identical, and the difference seems marginal. You certainly see an increase in the failure rate at the low-end of the rainfall distribution though (e.g. "Failure 3C" PDF larger than "Failure PD" PDF). And because heat is increasing you will still see increasingly frequent joint warm and dry conditions.

*The reviewer makes a good point, the sentence was badly written. It has been updated now:*

*Line 313: "... we see an increase in the failure rate at the low-end of the rainfall distribution for 2C and 3C ..."*

Figure 7: change the axis labels to be interpretable to the reader instead of being the variable names used in the code

*Thank you for the suggestion. We have updated the axis labels.*

L 282 -283 (and relevant to 390-392) - The interpretation that compound weather extremes become less important in a warming world because extreme temperature alone drives failures is not necessarily true. It may be (and probably is) rather that compound weather

extremes become more likely in your shuffled data when you uniformly warm all years. For example, if every year is an exceptionally hot year then your crop failures depend almost exclusively on rainfall, making the joint failure and shuffled failure years similar to one another at high return periods. In fact, we can see that all of the 3C failures still occur at very low levels of precipitation in Figure 7, but it is true that now low levels of precipitation lead to crop failures at higher rates.

*Thank you for the comment. This is true, there is a difference between the statistical 'compound factor' and the more physical 'compound event' that has led to confusion. We updated the term to 'relative compound contribution'. We also updated the text, explaining that the relative compound contribution is reducing, which means the correlation structure between the meteorological variables becomes less important, but that the compound conditions leading to crop failures are increasing due to higher temperatures during summer.*

*Line 305: "Thus, while the number of failure seasons increases with global warming for the original data, the increase in the number of failure seasons for the shuffled data is larger. A reduction in the relative compound contribution for the 2C and 3C scenarios suggests the correlation structure between meteorological variables becomes less relevant for our definition of crop failure."*

L 255 - I don't think the data support a robust increase of failure probability at higher precipitation levels unless you are willing to put equal weight on the corresponding dips in higher maximum temperatures leading to lower rates of failure and higher diurnal temperature ranges leading to lower rates of failure. I'd at least caution the reader that this may be noise rather than signal in the data.

*Thank you. This is correct, we have updated the text.*

*Line 279: "Precipitation shows a general inverse proportion to crop failure probability, suggesting low values of precipitation to increase failure probability, as indicated by Figure 4."*

L243 - define DTR at first use in the text. While it's defined in the table it is not defined before first use in the text

*Correct, it has been updated.*

L159-161 - what does assigning weights mean in this case? Was the dependent variable weighted? Are you running RF on binary data instead of crop yields generally? If so, justify this decision.

*The dependent variable has not been weighted. The weights are related to the penalty function associated with the random forest model. We run the RF on binary data, being divided into failure years (any year with yields below the historical standard deviation) and non-failure years (all the other years). This is a common approach, with previous works, such as Ben-Ari et al. (2018), Vogel et al. (2021) and Zhu et al. (2021), also following this approach.*

*Line 148: "Here we explore the probability of soybean failure, which is defined by means of a threshold, similarly to Ben-Ari et al. (2018); Vogel et al. (2021); Zhu et al. (2021). Every season with yield one standard deviation below the mean was considered a failure."*

- Also, why does it matter if the failure observations are less frequent?

*RF have a natural bias towards the majority class and suffer from performance issues when the dataset is imbalanced.*

*Line 173: " Because the crop yield dataset has less failure seasons than non-failure seasons, the dataset is imbalanced, affecting the model's capacity in identifying the minority class. To address this issue and improve the model performance, we assigned weights to the predictions of each class with values inversely proportional to their frequency. This increases the penalties for underrepresentation of the minority class, balancing the model."*

L158 - need to better define for the reader what a "data split" and a "shuffle" is in your cross validation procedure

- Described on L190

*Thank you for the comment. We added the following information to better explain the training and validation procedure:*

*Line 169: "We tuned the random forest's parameters following a resampling technique called cross-validation. It consisted of dividing the data in 10 different splits, where 9 splits are used to train the model and the remaining one is used for validation. The process is run 10 times so that every split is used once for testing. In addition this process was repeated 5 times with different random divisions, leading to 50 runs in total."*

L100 - note that soybean yield data is available for this entire period going back to 1900 from USDA, but it is difficult to remove the management and technology changes trend. So there is still reason to use crop models and the work provides a useful complement to observation-based analyses.

*Thank you for the comment. As this is related to the first minor comment, on the EPIC use of management and technology trends, we addressed in the same section:*

*Line 106: "Another advantage of using simulated crop models is that management and technology trends are static, whereas they are intrinsically embedded in the observed yields datasets"*

L 37: "the majority of climatic shocks are compound events" - is this true?

*Thank you for the comment. We tried to be more specific according to the referred literature as:*

*Line 41: "... the majority of climate-driven societal or natural shocks are the result of compound events (Zscheischler et al., 2017; Zampieri et al., 2017)"*

---

## Author Comment (AC2)

**Weather-induced crop failure events under climate change: a storyline approach**

Henrique M.D. Goulart, Karin van der Wiel, Christian Folberth, Juraj Balkovic, and Bart van den Hurk

**= Response to the reviewers =**

**Reviewer #2:**

Goulart et al. analyse weather conditions leading to soybean failure in the Midwest US using crop model data, training of a random forest model, and analysing a particular historical event and possible analogous events in the present day and future. The paper is well-written and clearly describes the approaches used and the results, and I enjoyed reading it. I have some comments particularly around the justification of the data used and the storyline approach, which I have noted below.

*We'd like to thank the reviewer for their positive feedback. We welcome the suggestions for the storyline section. Below we note the revisions done in response to all the suggestions.*

**Specific comments**

- I think the reasoning behind using a crop model rather than observations is sound, however you haven't shown whether the model is reliable at modelling soybean yields in the US. Could you provide any references, or analysis (if there is observed data available), to show whether the crop model provides realistic results for soybean production in the region of interest and can reproduce some of the important impacts being considered in the paper (e.g. does the model represent the impact of hot and dry conditions on soybean plants well?)? It would then be relevant to reflect in the discussion section on how the model-based results are likely to compare to real-world crop failures.

*Thank you for the very constructive comment. It is indeed necessary to contextualize the crop model performance with respect to the existing observed datasets. We added a comparison between the EPIC-IIASA simulated yields and the observed yield dataset of the USDA (United States Department of Agriculture) for the region considered. First, we detrended linearly the observed dataset, then we spatially averaged both datasets along the period of 1960 to 2016 and finally we standardized the two datasets, to focus on the interannual variability of the yield timeseries. We found that the $R^2$ between EPIC and the observed dataset is high, at 0.674, and that EPIC is capable of replicating the interannual variability of the observed data.*

*Line 115: "For validation of the crop model, we compared the EPIC-IIASA simulated yields with the observed yields from the US Department of Agriculture (USDA, www.nass.usda.gov/Quick_Stats) for the region considered. EPIC-IIASA has higher mean and standard deviations values than the observed as the simulated yields are potential (Folberth et al., 2016).To evaluate the interannual variability, we obtained a coefficient of determination, $R^2$, of 0.674. We also observed a good correlation between the two standardised datasets (Figure C1). We consider EPIC capable of replicating the interannual variability of the observed data."*

[Figure]

*Figure C1: Standardized comparison between the EPIC-IIASA simulated yields and the observed yield dataset of the USDA (United StatesDepartment of Agriculture) for the region considered in this work.*

- I assume all meteorological variables were considered at monthly scale for the analysis, but I don't think this is explicitly stated anywhere. Is there justification for using this timescale that could be provided in the text? Did you consider shorter timescale events that may also cause crops to fail but would be averaged out when looking at monthly timescales (e.g. a short very dry period), or if there are periods of less than a month in the crop lifecycle when the soybean may be more vulnerable to particular weather conditions?

*Thank you for the comment. We have modified the text to make this more explicit. We decided to work using the monthly scale because other works have shown that it is possible to find strong signals between climate and crop at a monthly scale (and therefore suppress some of the noise that arises when shorter time scales are used) (Ben-Ari et al., 2018; Vogel et al., 2021; Hamed et al., 2021). Nevertheless, we acknowledge increasing the time resolution could potentially provide more information and we updated the text to mention this.*

*Line 122: "... covers the period from 1901 to 2019 at a monthly scale ..."*

*Line 385: "The meteorological variables are at a monthly scale, which has been used in past studies as well (Ben-Ari et al., 2018; Vogel et al., 2021; Hamed et al., 2021.). However, adopting shorter timescales could lead to additional information on how weather interacts with crops."*

- Given that the paper is titled "a storyline approach", I think the storyline analysis is quite limited. For the 2012 season, only the meteorological variable values are presented and the probability of failure calculated. Can you include more insight into the storyline of events, e.g. how did the weather impact the crop via changes to the soil moisture? How did the high DTR affect the crop? This information would help to provide a much more complete chain of events for the 2012 season, otherwise it is unclear to me how it is a storyline approach being used to identify the drivers of the

event. I think using the impact analogues as well as the event analogues is a very interesting approach and, as you note in the discussion, the impact perspective is likely to be of more interest to society. However it would be interesting to include analysis of how the storyline chains of events differ (compared to the 2012 season and to each other) in these cases leading up to similar impacts. There is some relevant discussion already in section 4 but describing the range of plausible storylines more explicitly would help to incorporate more of a storyline approach in the paper.

*Thank you for the suggestion. We agree with the reviewer that the storyline part has been rather limited. Soil moisture is very relevant for agriculture studies, but here we focus only on creating connections between basic meteorological variables and soybean yields. Adding soil moisture, while relevant, would add extra layers of complication for data and analysis. Last, soil moisture is at least partially a compound product of temperature and precipitation fluctuations, which we already address in the analysis. We discuss the role of DTR in the discussion section, but have added more information to enrich this section. There are indeed different event cascades leading to 2012 analogues. The monthly temperature values of the 2012 analogues in a warmer climate (GW) are warmer than the original 2012 event, but even the normal years are already very close to the 2012 season; there is a slight drying trend during Jul-Aug in general for GW scenarios. The 2012 analogues are drier during Jul-Aug than the 2012 season, but also slightly wetter in May-June; DTR is decreasing with GW, and therefore the analogues do not reach the original 2012 season levels.*

*Line 365: " In addition, by relying on the impact metric, the meteorological conditions of the analogues can be analysed for changes due to global warming (Figures 10d, e, f and C10). The impact analogues of the 2012 year show warmer temperatures during summer with respect to the original event. For precipitation, the analogues are significantly drier than the 2012 year during July and August, the months in which the RF model takes into account. Finally, the analogues present lower DTR values during most of the year"*

*Line 450: "Furthermore, impact-analogues of the 2012 season in the future display a change in their physical properties: they are hotter and drier, but with lower values of diurnal temperature range."*

[Figure]

*Subtitle: Radar graphs showing the number of seasons exceeding the 2012 values for each meteorological variable for Present Day (a), 2 °C global warming scenario (b) and 3 °C global warming scenario (c) scenarios. Time series of the historical years, the 2012 season, the impact analogues and their corresponding climatology for maximum monthly temperature (d), precipitation (e) and diurnal temperature range (f).*

- In Figure 5a there is a pronounced jump in the observed data at return period of around 6 years. Are you able to provide any insight into why this might be?

    *Thank you for the comment. The jump we see around failure likelihood 0.5 is likely due to the way random forests work. They stipulate probabilities of failures for every single year, which in the end converge to either 1 (failures > 0.5) or 0 (failures < 0.5), and the resulting "S" shape is common, with most observations falling in either side of the threshold 0.5. For cases in which the sample is small, the jumps can appear, but for cases in which there are more samples, as it is the 2000-year samples we have for the GW scenarios, the whole distribution is properly fitted.*

**Technical corrections**

- Line 26: It seems strange to refer to Figure 2 in the text before Figure 1. Also this figure shows the mean yield rather than anything related to low yield in 2012. Combining information from Figure 9 might be relevant here to show the yield anomaly.

*We agree with the reviewer and decided to remove the first mention of Figure 2:*

- Figure 2: Where does this mean yield data come from (is it from the model or is it observed data)?

*It comes from the crop model, we added to the text:*

*Figure 2: "Selected grid points for the main producer states and the mean yields (ton/ha) per grid cell as simulated by the EPIC-IIASA model"*

- Line 28-30: Sentence starting "On a global level…" – I think there is a missing word or grammar check needed on this sentence

*We welcome the suggestion and have rewritten the sentence:*

*Line 32: "On a global level, interannual climate variability is responsible for approximately 30% of the year-to-year variability in crop yields (Lobell and Field, 2007) but the influence of interannual climate variability could rise up to 60% of the yield variability in certain regions (Ray et al., 2015; Frieler et al., 2017)."*

- Line 30: Change "Extremes" to "Extreme"

*Thank you for the correction. It has been updated.*

- Line 44-45: changing "to link" to "in linking" and "to explain" to "in explaining" would improve the readability here

*Thank you for the correction. It has been updated.*

- Line 166: Correct spelling of Matthews

*Thank you, it has been corrected.*

- Line 184: I assume RF is the abbreviation for Random Forest but this needs clarifying

*Thank you. We added the abbreviation for the first time Random Forest is mentioned:*

- Table C1: Need to define what the codes for the variables mean in this table or use their full names

*Thank you. We updated the label of the figure to mention the full names of the variables.*

[Figure]

**Figure C1.** The distribution of mean yearly temperatures in the Midwestern US of EC-Earth ensembles for Present Day scenario (blue), 2 °C global warming scenario (2C, orange) and 3 °C global warming scenario (3C, green) scenarios.

- Line 234: Do you mean high correlations with the yield or between the variables?

*We meant between the meteorological variables. We updated the text.*

*Line 259: "The remaining meteorological variables have high correlation levels between themselves"*

- Figure 4: It would be easier for the reader to interpret the figures if you could change the axis labels to the full names of the variables. Also it is not clear what the small lines along the x axis correspond to.

*Thank you for the comment. The other reviewers also questioned the purpose of the decile marks (ticks) and we decided to remove them, as they are not very informative. In addition, we added the full names of the variables as suggested.*

- Line 267: Is the observed data mentioned here the yield data from EPIC-IIASA, rather than real-world observations?

*The observed data has led to confusion, as it is the yield data from EPIC-IIASA. We updated the text as:*

*Line 291: "… is consistent with the historical data from EPIC-IIASA …"*

- Figure 9a): Are these yield anomalies from the model or is other observed data used?

*The yield anomalies correspond to the yield data from EPIC-IIASA. We updated the text as:*

*Figure 9: "… to the averaged historical yield data from EPIC-IIASA …"*

- Line 374: Do you mean Figure 10d here?

*Thank you for the correction, we updated the reference of the figure for the 10d.*

---

## Author Comment (AC3)

**Weather-induced crop failure events under climate change: a storyline approach**

Henrique M.D. Goulart, Karin van der Wiel, Christian Folberth, Juraj Balkovic, and Bart van den Hurk

**= Response to the reviewers =**

**Reviewer #3:**

Review of "Weather-induced crop failure events under climate change: a storyline approach"

This study explored climate impacts on soybean crop failures in the Midwestern United States under present and future warming scenarios using the random forest model and storyline approach with model simulated crop data and CRU climate data. The findings suggest that the failures are likely to increase with global warming, changes in both univariate and multivariate climate drivers of failure decrease the importance of compound factor, and impact-analogues are significantly increased under global warming compared to event analogues. The manuscript is well in line with the scope of the journal Earth System Dynamics. While the random forest model is not new, it is novel to use that to analyze climate impacts on crop failures. Also, it is very interesting to use storyline approach to study this issue. The findings are relevant. I think, in general, the paper is publishable after revisions. While the statistical analysis is done rigorously, the data may be improved, and method description and the interpretation of the results and findings also need to be improved. Here I provide specific comments as follows.

***We would like to thank the reviewer for their constructive feedback on our manuscript. We are grateful for the suggestions to improve our manuscript and we did our best efforts to address them in the comments below.***

Major comments:

1. It is reasonable to use model simulated crop data to obtain longer time series for model training. However, the study does not show the performance of the simulated data compared to observations. Since the study region has accessible observational yield data over long period, it should demonstrate the simulated crop data can well represent observations. Otherwise, it may require a bias correction to improve the quality of the simulated crop data.

   ***Thank you for the suggestion. This has been pointed out also by another reviewer. We added a comparison between the EPIC-IIASA simulated yields and the observed yield dataset of the USDA (United States Department of Agriculture) for the region considered. First, we detrended linearly the observed dataset, then we spatially averaged both datasets along the period of 1960 to 2016 and finally we standardized the two datasets, to focus on the interannual variability of the yield timeseries. We found that the $R^2$ between EPIC and the observed dataset is high, at 0.674, and that EPIC is capable of replicating the interannual variability of the observed data.***

   ***Line 115: "For validation of the crop model, we compared the EPIC-IIASA simulated yields with the observed yields from the US Department of Agriculture (USDA, www.nass.usda.gov/Quick_Stats) for the region***

*considered. EPIC-IIASA has higher mean and standard deviations values than the observed as the simulated yields are potential (Folberth et al., 2016).To evaluate the interannual variability, we obtained a coefficient of determination, $R^2$, of 0.674. We also observed a good correlation between the two standardised datasets (Figure C1). We consider EPIC capable of replicating the interannual variability of the observed data."*

[Figure]

*Figure C1: Standardized comparison between the EPIC-IIASA simulated yields and the observed yield dataset of the USDA (United StatesDepartment of Agriculture) for the region considered in this work.*

2.      It can be simple to average all the data over the region, but will discard many useful information. It is arbitrary to state "local scale of impacts are not meaningful for national and global implications". Using all the data points within the study region can increase the sample size by taking into account of the within region variability and increase the robustness of the model.

*Thank you for the comment. We agree with the reviewer that aggregating data spatially might lead to loss of information, especially on local extreme conditions. However, the aggregation allows us to prioritise the dominant meteorological dynamics in the region, improving the clarity of the main interactions between climate and crop. As described in Section 2.2, there were preliminary steps to clean the data and minimise the spatial variability to obtain a more clear signal when aggregating across the region studied. We selected only the main soybean producing states and only grids with rainfed soybeans, obtaining a sub-region of the Midwestern US. In addition, there are other works on climate-crop interactions that have used different techniques of spatial aggregation from sub-national areas to global areas, like Lobell and Field, 2007; Lesk et al., 2016; Heino et al., 2018 and Ben-Ari et al., 2018. Since we agree with the reviewer that there are potential drawbacks in aggregating gridded data, we added to the discussion section a clarification on the limitation of the approach.:*

*Line 140: "We spatially averaged all data for the region studied (Figure 2) to focus on the regional scale of weather events and their crop yield impacts, as these have larger influence on global markets. Aggregating data spatially might lead to loss of information, especially on local extreme conditions, but the RF model performance is comparable when running on aggregated data and on all grid points (Table C1).'*

*Line 388: "We spatially aggregate the climate and crop data over the region analysed to focus on crop failures and meteorological conditions at the regional scale. While this approach allows us to focus on the main dynamics of the region, information on local extreme conditions is not attained."*

3.      It is not clear how the random forest model was trained and validated. While a description of the evaluation metrics is provided, it is not clear how the data was split as training data and validation data. The validation data needs to be excluded as unseen data when is used for model training.

*We thank the reviewer for the suggestion. We added a sentence describing how the data was divided and how the model was trained and validated*

*Line 178: "With the RF setup complete, we trained and validated the RF model following the 80/20 split, where 80% of the data was used to train the model and the remaining 20% was used to validate the model's performance on unseen data."*

4.      All the results require thorough interpretations such as the mechanisms and biophysical meaning besides just reporting the data.

*Thank you for the comment. We believe the scope of the work is more focused on the climatological dynamics of global warming, crop failures and the societal implications that these might bring. Therefore, to explain in detail the biophysical mechanisms of soybean development is not the focus. We have added extra discussion points on DTR (which is answered in comment 5), and on the interpretation of the compound events associated to soybean failures:*

*Line 428: "From a multivariate perspective, the correlation structure of the variables contributes to the occurrence of compound events, as previously shown by (van den Hurk et al., 2015) and (Santos et al., 2021). Yet, we observe a decrease in its importance under global warming conditions for the failure yield threshold adopted here (Figure 6). A higher frequency of years with critical temperature during summer makes crop failures mostly dependent on precipitation values. Therefore, while still physically a compound event, the soybean failures under global warming become statistically similar to a univariate event based mostly on precipitation."*

5.      Based on the previous literature, the increase in DTR may actually benefit yields, which is contradictory with the finding from this study. Please explain.

*Thank you for the comment. Following the suggestion of the reviewer, we expanded on the literature about the influence of DTR on crop development. We found that DTR could be linked to both positive and negative impacts on crop yields, as high values of DTR could be a sign of increased heat or cold stress, and low DTR could be a consequence of high cloud coverage, low solar*

*radiation, or high rainfall. Therefore, we believe it is plausible that our statistical model found an association of crop failures with high values of DTR. We updated the text as:*

*Line 418: "The DTR projections indicate a descending trend, which in itself reduces crop failure probability according to our model. DTR is highly relevant for crop development, with previous studies showing the multiple impacts it can have on crops (Lobell, 2007; Zhang et al., 2013; Chen et al., 2015; Verón et al., 2015; Hernandez-Barrera et al., 2017; Rahman et al., 2017; van Etten et al., 2019). High values of DTR suggest peaks in high day-time temperature, which can disrupt the photosynthetic activity of crops (Allakhverdiev et al., 2008). High values of DTR can also indicate low night-time temperatures or night frosts, with capacity to damage crops during all stages of crop growth (Barlow et al., 2015). A study based on EPIC to simulate maize yields in the US has demonstrated that higher DTR values lead to greater evapotranspiration losses, reducing the yield outputs (Dhakhwa and Campbell, 1998). On the other hand, low values of DTR could indicate low solar radiation or high cloud coverage, both harmful for crops (van Etten et al., 2019; Vogel et al., 2019; Lobell, 2007). The decrease in DTR values due to global warming is mainly associated with a higher increase of minimum night-time temperature than daytime temperature (Qu et al., 2014; Sun et al., 2019). "*

Minor comments:

1. The approach formulated in this study consists of three parts. The storyline approach is part 3 of the approach (part c) and address part of the research objectives. However, based on the title, storyline approach seems the key message of this study. I think the title needs to be changed since the storyline approach seems not the major focus of this study.

    *While it is true that the last third of the work is the one dedicated to storylines, the chain of events that composes the storyline is developed in the first part (part a). We also believe the storylines are essential to the main message of this study: by creating a link between the meteorological conditions and the crop failure events, we were able to identify different meteorological conditions in the future projections that could lead to similar extreme-impact events to our case study. These different analogues following the climate-crop connection belong to the category of event-based storylines. Changing the title to "Storylines of weather-induced crop failure events under climate change" removes the focus on the storyline approach, but it expresses how we used storylines to create different analogues of a historical case.*

2. Little result is presented in the abstract. It can be more informative to shorten the background and methods descriptions and include more results.

    *We appreciate the suggestion. We added:*

    *Line 13: "We find that crop failures in the Midwestern US are linked to low precipitation levels, high temperature and diurnal temperature range (DTR) levels during July and August. Results suggest soybean failures are likely to increase with climate change. With more frequent warm years due to global warming, the joint hot-dry conditions leading to crop failures become mostly dependent on precipitation levels, reducing the importance of the relative*

*compound contribution. While event-analogues of the 2012 season are rare and not expected to increase, these analogues show a significant increase in occurrence frequency under global warming, but for different combinations of the meteorological drivers than experienced in 2012. This has implications for assessment of the drivers of extreme impact events."*

3.      Line 55: it is not clear to say "crop failures are compound events". Do you mean crop failures can be consequences of compound events? It requires a clear definition on compound events.

*Thank you for the correction. We mean crop failures are usually consequence of compound events. In line 36 we provide a definition of compound events. We updated the text as:*

*Line 59: "since crop failures are usually the result of compound meteorological drivers…"*

4.      Lines 74-75: "…, a natural next step is to …" It may be not so natural. I suggest the authors better explain their motivations and why such study is needed.

*Thank you for the comment. We updated the section with a clearer sentence on the motivation for this work (1ˢᵗ quote) and added a more explicit sentence on the research question (2ⁿᵈ quote).*

*Line 78: "Building on these works, we expand the studies of global warming impacts on agriculture to include multivariate analysis, by explicitly modelling the compound nature of meteorological variables and their interactions."*

*Line 80: "The aim of this work is  to understand how global warming affects the meteorological conditions leading to crop failures."*

5.      The last paragraph of the introduction mostly described the method/experimental outline. The majority of this part should be better move to the method section as a general statement of the method prior to detailed descriptions. The introduction section should focus on explaining why the research is needed, is novel, and stating the scope, research questions/objectives.

*Thank you for the comment. Here we follow the convention to conclude the introduction section with a brief reading guide, the outline.*

6.      Line 119: "… a similar 5-year period …" Please clarify what are the similar 5-year period at 2C and 3C.

*We updated the text to make the description clearer:*

*Line 131: "… a 5-year period representing an average global mean temperature 2°C above the pre-industrial levels (referred to as 2C) and another 5-year period corresponding to an average of 3°C above pre-industrial levels (referred to as 3C)"*

7.      Line 194: Please interpret what is compound factor besides providing the equation.

*Thank you for the comment. Another reviewer also had questions on the compound factor, and we decided to reformulate the compound factor to relative compound contribution, in order to avoid confusion with compound*

*events. The relative compound contribution quantifies how important the correlation structure of the meteorological variables is to crop failures, and how this might change with different levels of global warming. The interpretation of this metric is that at lower values of relative compound contribution, the order of the distribution of the meteorological variables becomes less important, which reduces the uncertainty of the future impacts with respect to the distribution of the meteorological variables. From a statistical point of view, it means the failure function becomes more dependent on a single variable, approximating to a univariate distribution. We updated the text as:*

*Line 209: "To assess the importance of the correlation of the conditions leading to failures, we created permuted versions of each large ensemble by randomly reshuffling the meteorological variables (van den Hurk et al., 2015; Santos et al.,2021), so that the correlation structure between them was removed (referred to as shuffled versions). We also defined a metric called the relative compound contribution. Relative compound contribution measures the importance of the correlation structure between the meteorological variables leading to crop failures. It is a statistical interpretation comparing crop failures under different correlation structures. Relative compound contribution is calculated as the ratio of the failure ratio obtained with the original data to the failure ratio obtained with the shuffled data. The closer to 1 the relative compound contribution gets, the less important the correlation structure between the variables is."*

8.    Lines 196-197: the definition of return period is confusing. Based on Figures 5 and 6, return period is not a one to one relationship with inverse of the failure probability, how it can be calculated as the inverse of the failure probability? It needs to be clarified.

*Thank you for showing us the ambiguity of the text. We corrected it. The return period is the inverse of the exceedance probability of an event to occur. In our study, we ordered the crop failure probabilities in an ascending order, calculated the exceedance probability and only then calculated the inverse of it to generate the return periods. So, the return period is not the inverse of the crop failure probability as the reviewer has correctly pointed out.*

*Line 219: "… comparing the failure probabilities for different return periods (calculated as the inverse of the exceedance probability of the specific event to occur) …"*

9.    It is not clear which month of data was used as candidate meteorological input prior to eliminating and combining monthly data in July and August.

*We added a figure in the SI to illustrate the R² scores of the meteorological variables grouped by month, and added a short description:*

*Line 248: "The considered soybean season in the US ranges from May to October, and we see that the months with the highest sensitivity to meteorological conditions are July and August (Figure C6)."*

[Figure]

10.     Variable names and acronyms are inconsistent in the text and tables and do not follow standard practice.

*Thank you for the suggestions. We updated the variable names and acronyms.*

11.     Figure 3b: Is Pearson's correlation performed on detrended or categorical yield?

*Thank you for the question. The Pearson's correlation was performed on continuous detrended yield datasets.*

12.     I would suggest only providing half of the correlation matrix. To limit redundancy

*Thank you for the suggestion, we updated the matrix.*

[Figure]

13.     Figure 4: I am assuming that the ticks on the bottom of the partial dependency plot represent the percentiles of the data. If they are being included, they need to be explained.

*Thank you for the comment. The other reviewers also questioned the purpose of the decile marks (ticks) and we decided to remove them, as they are not very informative. In addition, we added the full names of the variables as suggested.*

14.   Ln 390 decreasing compound factor. Is this just an artifact of choosing DTR.

*This is related to a comment by the other reviewer. The decreasing relative compound contribution is related to the increasing temperature levels under global warming scenarios. With more years warm, the crop failures become statistically more similar to a univariate distribution.*

*Line 428: "From a multivariate perspective, even though the correlation structure of the variables still contributes to compound events, as previously shown by (van den Hurket al., 2015) and (Santos et al., 2021), we observe a decrease in its importance under global warming conditions (Figure 6). A higher frequency of years with critical temperature during summer makes crop failures mostly dependent on precipitation values. Therefore, while still physically a compound event, the soybean failures under global warming become statistically similar to a univariate event based on precipitation"*

15.   Ln 374 Figure 10d?

*Thank you for the comment, it has been updated.*